

# 1 Reconstruction of the 1941 GLOF process chain at Lake Palcacocha (Cordillera Blanca, Perú)

*Martin Mergili[1,2], Shiva P. Pudasaini[3], Adam Emmer[4], Jan-Thomas Fischer[5], Alejo Cochachin[6], and Holger Frey[7]*
[1] Institute of Applied Geology, University of Natural Resources and Life Sciences (BOKU), Peter-Jordan-Straße 82, 1190 Vienna, Austria
[2] Geomorphological Systems and Risk Research, Department of Geography and Regional Research, University of Vienna, Universitätsstraße 7, 1010 Vienna, Austria
[3] Geophysics Section, Institute of Geosciences and Meteorology, University of Bonn, Meckenheimer Allee 176, 53115 Bonn, Germany
[4] Department of the Human Dimensions of Global Change, Global Change Research Institute, The Czech Academy of Sciences, Bělidla 986/4a, 603 00, Brno, Czech Republic
[5] Department of Natural Hazards, Austrian Research Centre for Forests (BFW), Rennweg 1, 6020 Innsbruck, Austria
[6] Unidad de Glaciología y Recursos Hídricos, Autoridad Nacional del Agua, Confraternidad Internacional 167, Huaráz, Perú
[7] Department of Geography, University of Zurich, Winterthurerstrasse 190, 8057 Zurich, Switzerland
Correspondence to: M. Mergili (martin.mergili@boku.ac.at)

## 18 Abstract

The Cordillera Blanca in Perú has been the scene of rapid deglaciation for many decades. One of numerous lakes
formed in the front of the retreating glaciers is the moraine-dammed Lake Palcacocha, which drained suddenly due to
an unknown cause in 1941. The resulting Glacial Lake Outburst Flood (GLOF) led to dam failure and complete drain-
age of Lake Jircacocha downstream, and to major destruction and thousands of fatalities in the city of Huaráz at a dis-
tance of 23 km. Lake Palcacocha has dramatically regrown through further glacial retreat since then and nowadays is
again considered a threat to the downstream communities. Previously, various types of computer simulations were
carried out in order to quantify the possible impact of future GLOFs from Lake Palcacocha, but no attempts are made
yet to back-calculate the 1941 event. Here, we chose an integrated approach to revisit the 1941 event in terms of topo-
graphic reconstruction and numerical back-calculation with the GIS-based open source mass flow/process chain simu-
lation framework r.avaflow. Thereby we consider four scenarios: (A) and (AX) breach of the moraine dam of Lake
Palcacocha due to retrogressive erosion, assuming two different fluid characteristics; (B) failure of the moraine dam
caused by the impact of a landslide onto the lake; and (C) geomechanical failure and collapse of the moraine dam. The
simulations largely yield empirically adequate results with physically plausible parameters, taking the documentation
of the 1941 event and previous calculations of future scenarios as reference. The results of the scenarios indicate that
the most likely initiation mechanism would be retrogressive erosion, possibly triggered by a minor impact wave
and/or facilitated by a weak stability condition of the moraine dam. However, the involvement of Lake Jircacocha
disguises part of the signal of process initiation farther downstream.
Keywords: GLOF, high-mountain lakes, Lake Palcacocha, numerical simulation, process chain, r.avaflow, two-phase
flows



## 1   Introduction

Glacial retreat in high-mountain areas often leads, after some lag time (Harrison et al., 2018), to the formation of pro-glacial lakes, which are impounded by moraine dams or bedrock swells. Such lakes may drain suddenly, releasing a large amount of water which may result in complex and potentially catastrophic process chains downstream. Glacial lakes and outburst floods (GLOFs) have been subject of numerous studies covering many mountain regions all around the globe (Hewitt, 1982; Haeberli, 1983; Richardson and Reynolds, 2000; Huggel et al., 2003; Breien et al., 2008; Hewitt and Liu, 2010; Bolch et al., 2011; Mergili and Schneider, 2011; Mergili et al., 2013; Clague and O'Connor, 2014; Emmer et al., 2015, 2016).

The Cordillera Blanca (Perú) represents the most glacierized mountain chain of the Tropics. Glacial lakes and GLOFs are particularly common there (Carey, 2005). 882 high-mountain lakes were identified by Emmer et al. (2016). Some of these lakes are susceptible to GLOFs (Vilímek et al., 2005; Emmer and Vilímek, 2013, 2014; ANA, 2014; Iturrizaga, 2014). A total of 28 geomorphologically effective GLOFs originating from moraine-dammed lakes have been documented (Emmer, 2017). Most recently, GLOFs were recorded at Lake Safuna Alta (2002 – the trigger was a rock avalanche into the lake; Hubbard et al., 2005), at Lake Palcacocha (2003 – landslide-induced overtopping of the dam; Vilímek et al., 2005), and at Lake 513 (2010 – triggered by an ice avalanche; Carey et al., 2012). Lake Artizón Alto was hit by a landslide from a moraine in 2012, which resulted in cascading effects involving three more lakes and entrainment of a considerable amount of debris in the Artizón Valley and, farther downstream, the Santa Cruz Valley (Mergili et al., 2018a). A pronounced peak in frequency of high-magnitude GLOFs, however, was already observed in the 1940s and 1950s, when lakes of notable size had formed behind steep terminal moraine walls (Emmer et al., 2019). The most prominent and well-documented GLOF in this period occurred on 13 December 1941, when Lake Palcacocha in the Quilcay Catchment drained suddenly, leading to a process chain that resulted in at least 1600 fatalities and major destruction in the town of Huaráz 23 km downstream (Broggi, 1942; Oppenheim, 1946; Concha, 1952; Wegner, 2014).

In the Cordillera Blanca, the local population is highly vulnerable to high-mountain process chains, often induced by GLOFs (Carey, 2005; Hofflinger et al., 2019). In order to mitigate this threat, tens of lakes in the Cordillera Blanca have been remediated through technical measures such as open cuts, artificial dams or tunnels during the last decades (Oppenheim, 1946; Zapata 1978; Portocarrero, 1984; Carey, 2005; Portocarrero, 2014; Emmer et al., 2018). However, the management of GLOF risk is a difficult task (Carey et al., 2014). Anticipation of GLOF cascades – and, as a consequence, also hazard mapping – relies to a large extent on the application of computational mass flow models (GAPHAZ, 2017). Important progress was made since the mid-20[th] Century: various models were developed, and have more recently been implemented in simulation software tools (Voellmy, 1955; Savage and Hutter, 1989; Iverson, 1997; Takahashi et al., 2002; Pitman and Le, 2005; McDougall and Hungr, 2004; Pudasaini and Hutter, 2007; Chisolm and McKinney, 2018). Most of these approaches represent single-phase mixture models. Tools like RAMMS (Christen et al., 2010) or FLO-2D were used for the simulation of GLOFs (Mergili et al., 2011). Schneider et al. (2014), Worni et al. (2014), and Somos-Valenzuela et al. (2016) have sequentially coupled two or more tools for simulating landslide – GLOF cascades. However, single-phase models do not describe the interactions between the solid and the fluid phase, or dynamic landslide-lake interactions, in an appropriate way, so that workarounds are necessary (Gabl et al., 2015). Worni et al. (2014) called for integrated approaches. They would have to build on two- or even three-phase models considering water, debris, and ice separately, but also the interactions between the phases and the flow transformations. Pudasaini (2012) introduced a general two-phase flow model considering mixtures of solid particles and viscous fluid which has been used for the simulation of computer-generated examples of sub-aqueous landslides and particle transport (Kafle et al., 2016, 2019) as well as GLOFs (Kattel et al., 2016).



The recently introduced open source GIS simulation framework r.avaflow (Mergili et al., 2017) applies an extended
version of the approach of Pudasaini (2012). It was used to back-calculate the 2012 Santa Cruz process chain involving
four lakes (Mergili et al., 2018a), and the 1962 and 1970 Huascarán landslides (Mergili et al., 2018b), both in the Cor-
dillera Blanca. These studies identified the capability of that tool to appropriately simulate the transformations at the
boundary of individual processes, where one process transforms to the next, as one of the major challenges. Open is-
sues include the proper understanding of wave generation as a response to landslides impacting high-mountain lakes
and, as a consequence, the quantification of essential parameters such as the volume of overtopping water and the
discharge (Westoby et al., 2014). Further, uncertainties in the model parameters and the initial conditions accumulate
at process boundaries (Schaub et al. 2016), and threshold effects are expected to result in strongly non-linear responses
of the model error (Mergili et al., 2018a, b). In high-energy mass flows, the physical characteristics of the processes
involved are not always understood at the required level of detail (Mergili et al., 2018b).
On the one hand, flow models and simulation tools can help us to better understand some of the key mechanisms of
high-mountain process chains. On the other hand, well documented case studies are important to gain a better under-
standing on which questions can be tackled with simulation tools, and which questions cannot be answered without
further research. In the present work, we explore this field of uncertainty by applying the r.avaflow computational
tool to the 1941 Lake Palcacocha GLOF process chain. Thereby, based on the simulation of different scenarios, we
investigate on the following research questions:

    1.  What is the most likely release mechanism of initiating the process chain of the 1941 GLOF of Lake Palcaco-
        cha?

    2.  Are we able to back-calculate this process chain in an empirically adequate way with physically plausible
        model parameters? Mergili et al. (2018b) reported a trade-off between these two criteria for the simulation of
        the 1970 Huascarán landslide.

    3.  What are the major challenges in achieving successful (empirically adequate and physically plausible) simula-
        tions?

    4.  What can we learn with regard to forward calculations of possible future events?

In Sect. 2 we depict the local conditions and the documentation of the event. After having introduced the computa-
tional framework r.avaflow (Sect. 3), we describe in detail the simulation input (Sect. 4) and our findings (Sect. 5). We
discuss the results (Sect. 6) and finally summarize the key points of the research (Sect. 7).

## 2  Lake Palcacocha

### 2.1  Quilcay catchment and Cojup Valley

Lake Palcacocha is part of a proglacial system in the headwaters of the Cojup Valley in the Cordillera Blanca, Perú
(Fig. 1). This system was – and is still – shaped by the glaciers originating from the southwestern slopes of Nevado
Palcaraju (6,264 m a.s.l.) and Nevado Pucaranra (6,156 m a.s.l.). A prominent horseshoe-shaped ridge of lateral and
terminal moraines marks the extent of the glacier during the first peak of the Little Ice Age, dated using lichenometry
to the 17[th] Century (Emmer, 2017). With glacier retreat, the depression behind the moraine ridge was filled with a
lake, named Lake Palcacocha. A photograph taken by Hans Kinzl in 1939 (Kinzl and Schneider, 1950) indicates a lake
level of 4,610 m a.s.l., allowing surficial outflow (Fig. 2a). Using this photograph, Vilímek et al. (2005) estimated a lake
volume between 9 and 11 million m³ at that time, whereas an unpublished estimate of the Autoridad Nacional del





Agua (ANA) arrived at approx. 13.1 million m³. It is assumed that the situation was essentially the same at the time of
the 1941 GLOF (Sect. 2.2).
The Cojup Valley is part of the Quilcay catchment, draining towards southwest to the city of Huaráz, capital of the
department of Ancash located at 3,090 m a.s.l. at the outlet to the Río Santa Valley (Callejon de Huaylas). The distance
between Lake Palcacocha and Huaráz is approx. 23 km, whereas the vertical drop is approx. 1,500 m. The Cojup Val-
ley forms a glacially shaped high-mountain valley in its upper part whilst cutting through the promontory of the Cor-
dillera Blanca in its lower part. 8 km downstream from Lake Palcacocha (15 km upstream of Huaráz), the landslide-
dammed Lake Jircacocha (4.8 million m³; Vilímek et al., 2005) existed until 1941 (Andres et al., 2018). The remnants
of this lake are still clearly visible in the landscape in 2017, mainly through the change in vegetation and the presence
of fine lake sediments (Fig. 2b). Table 1 summarizes the major characteristics of Lake Palcacocha and Lake Jircacocha
before the 1941 GLOF.
**2.2    1941 multi-lake outburst flood from Lake Palcacocha**
On 13 December 1941 part of the city of Huaráz was destroyed by a catastrophic GLOF-induced debris and mud flow,
with thousands of fatalities. Portocarrero (1984) gives a number of 4000 deaths, Wegner (2014) a number of 1800; but
this type of information has to be interpreted with care (Evans et al., 2009). The disaster was the result of a multi-lake
outburst flood in the upper part of the Cojup Valley. Sudden breach of the dam and the drainage of Lake Palcacocha
(Figs. 2c and e) led to a mass flow proceeding down the valley. Part of the eroded dam material, mostly coarse materi-
al, blocks and boulders, was deposited directly downstream from the moraine dam, forming an outwash fan typical for
moraine dam failures (Fig. 2c), whereas additional solid material forming the catastrophic mass flow was most likely
eroded further along the flow path (both lateral and basal erosion were observed; Wegner, 2014). The impact of the
flow on Lake Jircacocha led to overtopping and erosion of the landslide dam down to its base, leading to the complete
and permanent disappearance of this lake. The associated uptake of the additional water and debris increased the en-
ergy of the flow, and massive erosion occurred in the steeper downstream part of the valley, near the city of Huaráz.
Reports by the local communities indicate that the valley was deepened substantially, so that the traffic between vil-
lages was interrupted. According to Somos-Valenzuela et al. (2016), the valley bottom was lowered by as much as
50 m at some parts.
The impact area of the 1941 multi-GLOF and the condition of Lake Palcacocha after the event are well documented
through aerial imagery acquired in 1948 (Fig. 3). The image of Hans Kinzl acquired in 1939 (Fig. 2a) is the only record
of the status before the event. Additional information is available through eyewitness reports (Wegner, 2014). Howev-
er, as Lake Palcacocha is located in a remote, uninhabited area, no direct estimates of travel times or associated flow
velocities are available. Also the trigger of the sudden drainage of Lake Palcacocha remains unclear. Two mechanisms
appear most likely: (i) retrogressive erosion, possibly triggered by an impact wave related to calving or an ice ava-
lanche, resulting in overtopping of the dam (however, Vilímek et al., 2005 state that there are no indicators for such
an impact); or (ii) internal erosion of the dam through piping, leading to the failure.
**2.3    Lake evolution since 1941**
As shown on the aerial images from 1948, Lake Palcacocha was drastically reduced to a small remnant proglacial pond,
impounded by a basal moraine ridge within the former lake area, at a water level of 4563 m a.s.l., 47 m lower than
before the 1941 event (Fig. 3a). However, glacial retreat during the following decades led to an increase of the lake
area and volume (Vilímek et al., 2005). After reinforcement of the dam and the construction of an artificial drainage in
the early 1970s, a lake volume of 514,800 m³ was derived from bathymetric measurements (Ojeda, 1974). In 1974, two





artificial dams and a permanent drainage channel were installed, stabilizing the lake level with a freeboard of 7 m to
the dam crest (Portocarrero, 2014). By 2003, the volume had increased to 3.69 million m³ (Zapata et al., 2003). In the
same year, a landslide from the left lateral moraine caused a minor flood wave in the Cojup Valley (Fig. 2d). In 2016,
the lake volume had increased to 17.40 million m³ due to continued deglaciation (ANA, 2016). The potential of fur-
ther growth is limited since, as of 2017, Lake Palcacocha is only connected to a small regenerating glacier. Further, the
lake level is lowered artificially, using a set of siphons (it decreased by 3 m between December 2016 and July 2017).
Table 1 summarizes the major characteristics of Lake Palcacocha in 2016. The overall situation in July 2017 is illustrat-
ed in Fig. 2c.
## 2.4    Previous simulations of possible future GLOF process chains
Due to its history, recent growth, and catchment characteristics, Lake Palcacocha is considered hazardous for the
downstream communities, including the city of Huaráz (Fig. 2e). Whilst Vilímek et al (2005) point out that the lake
volume would not allow an event comparable to 1941, by 2016 the lake volume had become much larger than the
volume before 1941 (ANA, 2016). Even though the lower potential of dam erosion (Somos-Valenzuela et al., 2016) and
the non-existence of Lake Jircacocha make a 1941-magnitude event appear unlikely, the steep glacierized mountain
walls in the back of the lake may produce ice or rock-ice avalanches leading to impact waves, dam overtopping, ero-
sion, and subsequent mass flows. Investigations by Klimeš et al. (2016) of the steep lateral moraines surrounding the
lake indicate that failures and slides from moraines are possible at several sites, but do not have the potential to create
a major overtopping wave, partly due to the elongated shape of the lake. Rivas et al. (2015) elaborated on the possible
effects of moraine-failure induced impact waves. Recently, Somos-Valenzuela et al. (2016) have used a combination of
simulation approaches to assess the possible impact of process chains triggered by ice avalanching into Lake Palcaco-
cha on Huaráz. They considered three scenarios of ice avalanches detaching from the slope of Palcaraju (0.5, 1.0, and
3.0 million m³) in order to create flood intensity maps and to indicate travel times of the mass flow to various points of
interest. For the large scenario, the mass flow would reach the uppermost part of the city of Huaráz after approx.
1 h 20 min, for the other scenarios this time would increase to 2 h 50 min (medium scenario) and 8 h 40 min (small
scenario). Particularly for the large scenario, a high level of hazard is identified for a considerable zone near the Quil-
cay River, whereas zones of medium or low hazard become more abundant with the medium and small scenarios, or
with the assumption of a lowered lake level (Somos-Valenzuela et al., 2016). In addition, Chisolm and McKinney
(2018) analyzed the dynamics impulse waves generated by avalanches using FLOW-3D. A similar modelling approach
was applied by Frey et al. (2018) to derive a map of GLOF hazard for the Quilcay catchment. For Lake Palcacocha the
same ice avalanche scenarios as applied by Somos-Valenzuela et al. (2016) were employed, with correspondingly com-
parable results in the Cojup Valley and for the city of Huaráz.
# 3    The r.avaflow computational tool
r.avaflow is an open source tool for simulating the dynamics of complex mass flows in mountain areas. It employs a
two-phase model including solid particles and viscous fluid, making a difference to most other mass flow simulation
tools which build on one-phase mixture models. r.avaflow considers the interactions between the phases as well as
erosion and entrainment of material from the basal surface. Consequently, it is well-suited for the simulation of com-
plex, cascading flow-type landslide processes. The r.avaflow framework is introduced in detail by Mergili et al. (2017),
only those aspects relevant for the present work are explained here.
The Pudasaini (2012) two-phase flow model is used for propagating mass flows from at least one defined release area
through a Digital Terrain Model (DTM). Flow dynamics is computed through depth-averaged equations describing the





conservation of mass and momentum for both solid and fluid. The solid stress is computed on the basis of the Mohr-
Coulomb plasticity, whereas the fluid is treated with a solid-volume-fraction-gradient-enhanced non-Newtonian vis-
cous stress. Virtual mass due to the relative motion and acceleration, and generalized viscous drag, account for the
strong transfer of momentum between the phases. Also buoyancy is considered. The momentum transfer results in
simultaneous deformation, separation, and mixing of the phases (Mergili et al., 2018a). Pudasaini (2012) gives a full
description of the set of equations.
Certain enhancements are included, compared to the original model: for example, drag and virtual mass are computed
according to extended analytical functions constructed by Pudasaini (2019a, b). Additional (complementary) function-
alities include surface control, diffusion control, and basal entrainment (Mergili et al., 2017, 2018a, 2019). A conceptu-
al model is used for entrainment: thereby, the empirically derived entrainment coefficient $C_E$ is multiplied with the
flow kinetic energy:
$$q_{E,s} = C_E \left| T_s + T_f \right| \alpha_{s,E}, \; q_{E,f} = C_E \left| T_s + T_f \right| \left( 1 - \alpha_{s,E} \right). \quad (1)$$
$q_{E,s}$ and $q_{E,f}$ (m s$^{-1}$) are the solid and fluid entrainment rates, $T_s$ and $T_f$ (J) are the solid and fluid kinetic energies, and $\alpha_{s,E}$
is the solid fraction of the entrainable material (Mergili et al., 2019). Flow heights and momenta as well as the change
of elevation of the basal surface are updated at each time step (Mergili et al., 2017).
Any desired combination of solid and fluid release and entrainable heights can be defined. The main results are raster
maps of the evolution of solid and fluid flow heights, velocities, and entrained heights in time. Pressures and kinetic
energies are derived from the flow heights and velocities. Output hydrographs can be generated as an additional op-
tion (Mergili et al., 2018a). Spatial discretization works on the basis of GIS raster cells: the flow propagates between
neighbouring cells during each time step. The Total Variation Diminishing Non-Oscillatory Central Differencing
(TVD-NOC) Scheme (Nessyahu and Tadmor, 1990; Tai et al., 2002; Wang et al., 2004) is employed for solving the
model equations. This approach builds on a staggered grid, in which the system is shifted half the cell size during each
step in time (Mergili et al., 2018b).
r.avaflow operates as a raster module of the open source software GRASS GIS 7 (GRASS Development Team, 2019),
employing the programming languages Python and C as well as the R software (R Core Team, 2019). More details
about r.avaflow are provided by Mergili et al. (2017).

## 4   Simulation input

The simulations build on the topography, represented by a DTM, and on particular sets of initial conditions and model
parameters. For the DTM, we use a 5 m resolution Digital Elevation Model provided by the Peruvian Ministry of Envi-
ronment, MINAM (Horizons, 2013). It was deduced from recent stereo aerial photographs and airborne LiDAR. The
DEM is processed in order to derive a DTM representing the situation before the 1941 event. Thereby, we neglect the
possible error introduced by the effects of vegetation or buildings, and focus on the effects of the lakes and of erosion
(Fig. 4):
1.   For the area of Lake Palcacocha the elevation of the lake surface is replaced by a DTM of the lake bathymetry
232           derived from ANA (2016). Possible sedimentation since that time is neglected. The photograph of Hans Kinzl
233           from 1939 (Fig. 2a) is used to reconstruct the moraine dam before the breach, and the glacier at the same time.
234           As an exact positioning of the glacier terminus is not possible purely based on the photo, the position is opti-
235           mized towards a lake volume of approx. 13 million m³, following the estimate of ANA. It is further assumed





that there was surficial drainage of Lake Palcacocha as suggested by Fig. 2a, i.e. the lowest part of the moraine crest is set equal to the former lake level of 4,610 m a.s.l (Fig. 4b).

2. Also for Lake Jircacocha, surficial overflow is assumed (a situation that is observed for most of the recent landslide-dammed lakes in the Cordillera Blanca). On this basis the landslide dam before its breach is reconstructed, guided by topographic and geometric considerations. The lowest point of the dam crest is set to 4,130 m a.s.l. (Fig. 4c).

3. Erosional features along the flow channel are assumed to largely relate to the 1941 event. These features are filled accordingly (see Table 2 for the filled volumes). In particular, the flow channel in the lower part of the valley, reportedly deepened by up to 50 m in the 1941 event (Vilímek et al., 2005), was filled in order to represent the situation before the event in a plausible way (Fig. 4d).

All lakes are considered as fluid release volumes in r.avaflow. The initial level of Lake Palcacocha in 1941 is set to 4,610 m a.s.l., whereas the level of Lake Jircacocha is set to 4,129 m a.s.l. The frontal part of the moraine dam impounding Lake Palcacocha and the landslide dam impounding Lake Jircacocha are considered as entrainable volumes. Further, those areas filled up along the flow path (Fig. 4d) are considered entrainable, mainly following Vilímek et al. (2005). However, as it is assumed that part of the material was removed through secondary processes or afterwards, only 75% of the added material are allowed to be entrained. All entrained material is considered 80% solid and 20% fluid per volume.

The reconstructed lake, breach, and entrainable volumes are shown in Tables 1 and 2. The glacier terminus in 1941 was located in an area where the lake depth increases by several tens of metres, so that small misestimates in the position of the glacier tongue may result in large misestimates of the volume, so that some uncertainty has to be accepted.

We consider four scenarios:

A     Retrogressive erosion, possibly induced by minor or moderate overtopping. This scenario is related to a possible minor impact wave, caused for example by calving of ice from the glacier front, an increased lake level due to meteorological reasons, or a combination of these factors. In the simulation, the process chain is started by cutting an initial breach into the dam in order to initiate overtopping and erosion. The fluid phase is considered as pure water.

AX     Similar to Scenario A, but with the second phase considered a mixture of fine mud and water. For this purpose, density is increased to 1,100 instead of 1,000 kg m⁻³, and a yield strength of 5 Pa is introduced (Table 3). For simplicity, we still refer to this mixture as a fluid. Such changed phase characteristics may be related to the input of fine sediment into the lake water (e.g. caused by a landslide from the lateral moraine as triggering event), but are mainly considered here in order to highlight the effects of uncertainties in the definition and parameterization of the two-phase mixture flow.

B     Retrogressive erosion, induced by violent overtopping. This scenario is related to a large impact wave caused by a major rock/ice avalanche or ice avalanche rushing into the lake. In the simulation, the process chain is initiated through a hypothetic landslide of 3 million m³ of 75% solid and 25% fluid material, following the large scenario of Somos-Valenzuela et al. (2016) in terms of volume and release area. In order to be consistent with Scenario A, fluid is considered as pure water.

C     Internal erosion-induced failure of the moraine dam. Here, the process chain is induced by the collapse of the entire reconstructed breach volume (Fig. 4b). In the simulation, this is done by considering this part of the moraine not as entrainable volume, but as release volume (80% solid, 20% fluid, whereby fluid is again considered as pure water).





Failure of the dam of Lake Jircacocha is assumed having occurred through overtopping and retrogressive erosion, in-
duced by the increased lake level and a minor impact wave from the flood upstream. No further assumptions of the
initial conditions are required in this case.
The model parameter values are selected in accordance with experiences gained from previous simulations with
r.avaflow for other study areas, and are summarized in Table 3. Three parameters mainly characterizing the flow fric-
tion (basal friction of solid $\delta$, ambient drag coefficient $C_{AD}$, and fluid friction coefficient $C_{FF}$) and the entrainment coef-
ficient $C_E$ are optimized in a spatially differentiated way to maximize the empirical adequacy of the simulations in
terms of estimates of impact areas, erosion depths, and flow and breach volumes. As no travel times or velocities are
documented for the 1941 event, we use the values given by Somos-Valenzuela et al. (2016) as a rough reference. Vary-
ing those four parameters while keeping the others constant helps us to capture variability while minimizing the de-
grees of freedom, remaining aware of possible equifinality issues (Beven, 1996; Beven and Freer, 2001).
A particularly uncertain parameter is the empirical entrainment coefficient $C_E$ (Eq. 1). In order to optimize $C_E$, we
consider (i) successful prediction of the reconstructed breach volumes; and (ii) correspondence of peak discharge with
published empirical equations on the relation between peak discharge, and lake volume and dam height (Walder and
O'Connor, 1997). Table 4 summarizes these equations for moraine dams (applied to Lake Palcacocha) and landslide
dams (applied to Lake Jircacocha), and the values obtained for the regression and the envelope, using the volumes of
both lakes. We note that Table 4 reveals very large differences – roughly one order of magnitude – between regression
and envelope. In case of the breach of the moraine dam of Lake Palcacocha, we consider an extreme event due to the
steep, poorly consolidated, and maybe soaked moraine, with a peak discharge close to the envelope (approx..
15,000 m³ s⁻¹). For Lake Jircacocha, in contrast, the envelope values of peak discharge do not appear realistic. However,
due to the high rate of water inflow from above, a value well above the regression line still appears plausible, even
though the usefulness of the empirical laws for this type of lake drainage can be questioned. The value of $C_E$ optimized
for the dam of Lake Jircacocha is also used for entrainment along the flow path.
All of the computational experiments are run with 10 m spatial resolution. Only flow heights ≥25 cm are considered
for visualization and evaluation. We now describe one representative simulation result for each of the considered sce-
narios, thereby spanning the most plausible and empirically adequate field of simulations.

## 5   r.avaflow simulation results

### 5.1   Scenario A – Event induced by overtopping; fluid without yield strength

Outflow from Lake Palcacocha starts immediately, leading to (1) lowering of the lake level and (2) retrogressive ero-
sion of the moraine dam. The bell-shaped fluid discharge curve at the hydrograph profile O1 (Fig. 4) reaches its peak
of 18,700 m³ s⁻¹ after approx. 780 s, and then decreases to a small residual (Fig. 5a). Channel incision happens quickly –
53 m of lowering of the terrain at the reference point R1 occurs in the first less than 1200 s, whereas the lowering at
the end of the simulation is 60 m (Fig. 6a). This number represents an underestimation, compared to the reference
value of 76 m (Table 2). The lake level decreases by 42 m, whereby 36.5 m of the decrease occur within the first
1200 s. The slight underestimation, compared to the reference value of 47 m of lake level decrease, is most likely a
consequence of uncertainties in the topographic reconstruction. A total amount of 1.5 million m³ is eroded from the
moraine dam of Lake Palcacocha, corresponding to an underestimation of 22%, compared to the reconstructed breach
volume. Underestimations mainly occur at both sides of the lateral parts of the eroded channel near the moraine crest
– an area where additional post-event erosion can be expected, so that the patterns and degree of underestimation
appear plausible (Fig. 7a). In contrast, some overestimation of erosion occurs in the inner part of the dam. For numeri-

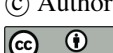



cal reasons, some minor erosion is also simulated away from the eroded channel. The iterative optimization procedure
results in an entrainment coefficient $C_E = 10^{-6.75}$.
The deposit of much of the solid material eroded from the moraine dam directly downstream from Lake Palcacocha, as
observed in the field (Fig. 2c), is reasonably well reproduced by this simulation, so that the flow proceeding down-
valley is dominated by the fluid phase (Fig. 8). It reaches Lake Jircacocha after $t = 840$ s (Fig. 5b). As the inflow occurs
smoothly, there is no impact wave in the strict sense, but it is rather the steadily rising water level (see Fig. 6b for the
evolution of the water level at the reference point R2) inducing overtopping and erosion of the dam. This only starts
gradually after some lag time, at approx. $t = 1,200$ s. The discharge curve at the profile O2 (Fig. 4) reaches its pro-
nounced peak of 750 m³ s⁻¹ solid and 14,700 m³ s⁻¹ fluid material at $t = 2,340$ s, and then tails off slowly.
In the case of Lake Jircacocha, the simulated breach is clearly shifted south, compared to the observed breach. With
the optimized value of the entrainment coefficient $C_E = 10^{-7.15}$, the breach volume is underestimated by 24%, compared
to the reconstruction (Fig. 7b). Also here, this intentionally introduced discrepancy accounts for some post-event ero-
sion. However, we note that volumes are uncertain as the reconstruction of the dam of Lake Jircacocha – in contrast to
Lake Palcacocha – is a rough estimation due to lacking reference data.
Due to erosion of the dam of Lake Jircacocha, and also erosion of the valley bottom and slopes, the solid fraction of the
flow increases considerably downstream. Much of the solid material, however, is deposited in the lateral parts of the
flow channel, so that the flow arriving at Huaráz is fluid-dominated again (Fig. 8). The front enters the alluvial fan of
Huaráz at $t = 2,760$ s, whereas the broad peak of 10,500 m³ s⁻¹ of fluid and 2,000 m³ s⁻¹ of solid material (solid fraction
of 16%) is reached in the period between 3,600 and 3,780 s (Fig. 4; Fig. 5c). Discharge decreases steadily afterwards. A
total of 2.5 million m³ of solid and 14.0 million m³ of fluid material pass the hydrograph profile O3 until t = 5,400 s.
Referring only to the solid, this is less material than reported by Kaser and Georges (2003). However, (i) there is still
some material coming after, and (ii) pore volume has to be added to the solid volume, so that the order of magnitude
of material delivered to Huaráz corresponds to the documentation in a better way. Still, the solid ratio of the hydro-
graph might represent an underestimation.
As prescribed by the parameter optimization, the volumes entrained along the channel are in the same order of mag-
nitude, but lower than the reconstructed volumes summarized in Table 2: 0.7 million m³ of material are entrained
upstream and 1.5 million m³ downstream of Lake Jircacocha, and 5.3 million m³ in the promontory. Fig. 9a summariz-
es the travel times and the flow velocities of the entire process chain. Frontal velocities mostly vary between 5 m s⁻¹
and 20 m s⁻¹, with the higher values in the steeper part below Lake Jircacocha. The low and undefined velocities di-
rectly downstream of Lake Jircacocha reflect the time lag of substantial overtopping. The key numbers in terms of
times, discharges, and volumes are summarized in Table 5.
**5.2    Scenario AX – Event induced by overtopping; fluid with yield strength**
Adding a yield strength of $\tau_y = 5$ Pa to the characteristics of the fluid substantially changes the temporal rather than
the spatial evolution of the process cascade. As the fluid now behaves as fine mud instead of water and is more re-
sistant to motion, velocities are lower, travel times are much longer, and the entrained volumes are smaller than in the
Scenario A (Fig. 9b; Table 5). The peak discharge at the outlet of Lake Palcacocha is reached at $t = 1,800$ s. Fluid peak
discharge of 8,200 m³ s⁻¹ is less than half the value yielded in Scenario A (Fig. 5d). The volume of material eroded from
the dam is only slightly smaller than in Scenario A (1.4 versus 1.5 million m³). The numerically induced false positives
with regard to erosion observed in Scenario A are not observed in Scenario AX, as the resistance to oscillations in the
lake is lower with the added yield strength (Fig. 7c). Still, the major patterns of erosion and entrainment are the same.
Interestingly, erosion is deeper in Scenario AX, reaching 76 m at the end of the simulation (Fig. 6c) and therefore the





base of the entrainable material (Table 2). This is most likely a consequence of the spatially more concentrated flow
and therefore higher erosion rates along the centre of the breach channel, with less lateral spreading than in Scenar-
io A.
Consequently, also Lake Jircacocha is reached later than in Scenario A (Fig. 6d), and the peak discharge at its outlet is
delayed (t = 4,320 s) and lower (7,600 m³ s$^{-1}$ of fluid and 320 m³ s$^{-1}$ of solid material) (Fig. 5e). 2.0 million m³ of materi-
al are entrained from the dam of Lake Jircacocha, with similar spatial patterns as in Scenario A (Fig. 7d). Huaráz is
reached after $t$ = 4,200 s, and the peak discharge of 5,000 m³ s$^{-1}$ of fluid and 640 m³ s$^{-1}$ of solid material at O3 occurs
after $t$ = 6,480 s (Fig. 5f). This corresponds to a solid ratio of 11%. Interpretation of the solid ratio requires care here as
the fluid is defined as fine mud, so that the water content is much lower than the remaining 89%. The volumes en-
trained along the flow channel are similar in magnitude to those obtained in the simulation of Scenario A (Table 5).
**5.3    Scenario B – Event induced by impact wave**
Scenario B is based on the assumption of an impact wave from a 3 million m³ landslide. However, due to the relatively
gently-sloped glacier tongue heading into Lake Palcacocha at the time of the 1941 event (Figs. 2a and 4b), only a small
fraction of the initial landslide volume reaches the lake, and impact velocities and energies are reduced, compared to a
direct impact from the steep slope. Approx. 1 million m³ of the landslide have entered the lake until $t$ = 120 s, an
amount which only slightly increases thereafter. Most of the landslide deposits on the glacier surface. Caused by the
impact wave, discharge at the outlet of Lake Palcacocha (O1) sets on at $t$ = 95 s and, due to overtopping of the impact
wave, immediately reaches a relatively moderate first peak of 7,000 m³ s$^{-1}$ of fluid discharge. The main peak of
16,900 m³ s$^{-1}$ of fluid and 2,000 m³ s$^{-1}$ of solid discharge occurs at $t$ = 1,200 s due do the erosion of the breach channel.
Afterwards, discharge decreases relatively quickly to a low base level (Fig. 10a). The optimized value of $C_E$ = 10$^{-6.75}$ is
used also for this scenario. The depth of erosion along the main path of the breach channel is clearly less than in the
Scenario A (Fig. 6e). However, Table 5 shows a higher volume of eroded dam material than the other scenarios. These
two contradicting patterns are explained by Fig. 11a: the overtopping due to the impact wave does not only initiate
erosion of the main breach, but also of a secondary breach farther north. Consequently, discharge is split among the
two breaches and therefore less concentrated, explaining the lower erosion at the main channel despite a larger total
amount of eroded material. The secondary drainage channel can also be deduced from observations (Fig. 3a), but has
probably played a less important role than suggested by this simulation.
The downstream results of Scenario B largely correspond to the results of the Scenario A, with some delay partly relat-
ed to the time from the initial landslide to the overtopping of the impact wave. Discharge at the outlet of Lake Jircaco-
cha peaks at $t$ = 2,700 s (Fig. 10b), and the alluvial fan of Huaráz is reached after 3,060 s (Fig. 10c). The peak discharges
at O2 and O3 are similar to those obtained in the Scenario A. The erosion patterns at the dam of Lake Jircacocha
(again, $C_E$ = 10$^{-7.15}$) very much resemble those yielded with the scenarios A and AX (Fig. 11b), and so does the volume
of entrained dam material (2.2 million m³). The same is true for the 2.5 million m³ of solid and 13.9 million m³ of fluid
material entering the area of Huaráz until $t$ = 5,400 s, according to this simulation.
Also in this scenario, the volumes entrained along the flow channel are very similar to those obtained in the simula-
tion of Scenario A. The travel times and frontal velocities – resembling the patterns obtained in Scenario A, with the
exception of the delay – are shown in Fig. 12a, whereas Table 5 summarizes the key numbers in terms of times, vol-
umes, and discharges.





### 5.4 Scenario C – Event induced by dam collapse

In Scenario C, we assume that the breached part of the moraine dam collapses, the collapsed mass mixes with the water from the suddenly draining lake, and flows downstream. The more sudden and powerful release, compared to the two other scenarios, leads to higher frontal velocities and shorter travel times (Fig. 12b; Table 5).

In contrast to the other scenarios, impact downstream starts earlier, as more material is released at once, instead of steadily increasing retrogressive erosion and lowering of the lake level. The fluid discharge at O1 peaks at almost 40,000 m³ s$^{-1}$ (Fig. 10d) rapidly after release. Consequently, Lake Jircacocha is reached already after 720 s, and the impact wave in the lake evolves more quickly than in all the other scenarios considered (Fig. 6f). The lake drains with a peak discharge of 15,400 m³ s$^{-1}$ of fluid and 830 m³ s$^{-1}$ of solid material after 1,680–1,740 s (Fig. 10e). In contrast to the more rapid evolution of the process chain, discharge magnitudes are largely comparable to those obtained with the other scenarios. The same is true for the hydrograph profile O3: the flow reaches the alluvial fan of Huaráz after $t = 2,160$ s, with a peak discharge slightly exceeding 10,000 m³ s$^{-1}$ of fluid and 2,000 m³ s$^{-1}$ of solid material between $t = 2,940$ s and 3,240 s. 2.7 million m³ of solid and 14.6 million m³ of fluid material enter the area of Huaráz until $t = 5,400$ s, which is slightly more than in the other scenarios, indicating the more powerful dynamics of the flow (Table 5). The fraction of solid material arriving at Huaráz remains low, with 16% solid at peak discharge and 15% in total. Again, the volumes entrained along the flow channel are very similar to those obtained with the simulations of the other scenarios (Table 5).

## 6 Discussion

### 6.1 Possible trigger of the GLOF process chain

There is disagreement upon the trigger of the 1941 multi-lake outburst flood in the Quilcay catchment. Whereas, according to contemporary reports, there is no evidence of a landslide (for example, ice avalanche) impact onto the lake (Vilímek et al., 2005; Wegner, 2014), and dam rupture would have been triggered by internal erosion, some authors postulate an at least small impact starting the process chain (Portocarrero, 2014; Somos-Valenzuela et al., 2016).

Each of the three assumed initiation mechanisms of the 1941 event, represented by the Scenarios A/AX, B, and C, yields results which are plausible in principle. We consider a combination of all three mechanisms a likely cause of this extreme process chain. Overtopping of the moraine dam, possibly related to a minor impact wave, leads to the best correspondence of the model results with the observation, documentation, and reconstruction. Particularly the signs of minor erosion of the moraine dam north of the main breach (Fig. 3a) support this conclusion: a major impact wave, resulting in violent overtopping of the entire frontal part of the moraine dam, would supposedly also have led to more pronounced erosion in that area, as to some extent predicted by the Scenario B. There is also no evidence for strong landslide-glacier interactions (massive entrainment of ice or even detachment of the glacier tongue) which would be likely scenarios in the case of a very large landslide. Anyway, the observations do not allow for substantial conclusions on the volume of a hypothetic triggering landslide: as suggested by Scenario B, even a large landslide from the slopes of Palcaraju or Pucaranra could have been partly alleviated on the rather gently sloped glacier tongue between the likely release area and Lake Palcacocha.

The minor erosional feature north of the main breach was already visible in the photo of Kinzl (Fig. 2a), possibly indicating an earlier, small GLOF. It remains unclear whether it was reactivated in 1941. Such a reactivation could only be directly explained by an impact wave, but not by retrogressive erosion only (A/AX) or internal failure of the dam (C) – so, more research is needed here. The source area of a possible impacting landslide could have been the slopes of Pal-





caraju or Pucaranra (Fig. 1), or the calving glacier front (Fig. 2a). Attempts to quantify the most likely release volume and material composition would be considered speculative due to the remaining difficulties in adequately simulating landslide-(glacier-)lake interactions (Westoby et al., 2014). Further research is necessary in this direction. In any case, a poor stability condition of the dam (factor of safety ∼ 1) could have facilitated the major retrogressive erosion of the main breach. A better understanding of the hydro-mechanical load applied by a possible overtopping wave and the mechanical strength of the moraine dam could help to resolve this issue.

The downstream patterns of the flow are largely similar for each of the scenarios A, AX, B, and C, with the exception of travel times and velocities. Interaction with Lake Jircacocha disguises much of the signal of process initiation. Lag times between the impact of the flow front on Lake Jircacocha and the onset of substantial overtopping and erosion are approx. 10 minutes in the scenarios A and B, and less than 3 minutes in the Scenario C. This clearly reflects the slow and steady onset of those flows generated through retrogressive erosion. The moderate initial overtopping in Scenario B seems to alleviate before reaching Lake Jircacocha. Sudden mechanical failure of the dam (Scenario C), in contrast, leads to a more sudden evolution of the flow, with more immediate downstream consequences.

## 6.2 Parameter uncertainties

We have tried to back-calculate the 1941 event in a way reasonably corresponding to the observation, documentation and reconstruction, and building on physically plausible parameter sets. Earlier work on the Huascarán landslides of 1962 and 1970 has demonstrated that empirically adequate back-calculations are not necessarily plausible with regard to parameterization (Mergili et al., 2018b). This issue may be connected to equifinality issues (Beven, 1996; Beven and Freer, 2001), and in the case of the very extreme and complex Huascarán 1970 event, by the inability of the flow model and its numerical solution to adequately reproduce some of the process components (Mergili et al., 2018b). In the present work, however, reasonable levels of empirical adequacy and physical plausibility are achieved. Open questions remain with regard to the spatial differentiation of the basal friction angle required to obtain adequate results (Table 3): lower values of $\delta$ downstream from the dam of Lake Jircacocha are necessary to ensure that a certain fraction of solid passes the hydrograph profile O3 and reaches Huaráz. Still, solid fractions at O3 appear rather low in all simulations. A better understanding of the interplay between friction, drag, virtual mass, entrainment, deposition, and phase separation could help to resolve this issue (Pudasaini and Fischer, 2016a, b; Pudasaini, 2019a, b).

The empirically adequate reproduction of the documented spatial patterns is only one part of the story (Mergili et al., 2018a). The dynamic flow characteristics (velocities, travel times, hydrographs) are commonly much less well documented, particularly for events in remote areas which happened a long time ago. Therefore, direct references for evaluating the empirical adequacy of the dimension of time in the simulation results are lacking. However, travel times play a crucial role related to the planning and design of (early) warning systems and risk reduction measures (Hofflinger et al., 2019). Comparison of the results of the scenarios A and AX (Fig. 9) reveals almost doubling travel times when adding a yield stress to the fluid fraction. In both scenarios, the travel times to Huaráz are within the same order of magnitude as the travel times simulated by Somos-Valenzuela et al. (2016) and therefore considered plausible, so that it is hard to decide about the more adequate assumption. Even though the strategy of using the results of earlier simulations as reference may increase the robustness of model results, it might also reproduce errors and inaccuracies of earlier simulation attempts, and thereby confirm wrong results.

The large amount of more or less pure lake water would point towards the Scenario A, whereas intense mixing and entrainment of fine material would favour the Scenario AX. More work is necessary in this direction, also considering possible phase transformations (Pudasaini and Krautblatter, 2014). At the same time, the optimization and evaluation



of the simulated discharges remains a challenge. Here we rely on empirical relationships gained from the analysis of
comparable events (Walder and O'Connor, 1997).

### 6.3 Implications for predictive simulations

Considering what was said above, the findings from the back-calculation of the 1941 event can help us to better un-
derstand and constrain possible mechanisms of this extreme process chain. However, they should only be applied for
forward simulations in the same area or other areas with utmost care. The initial conditions and model parameters are
not necessarily valid for events of different characteristics and magnitudes (Mergili et al., 2018b). In the case of Lake
Palcacocha, the situation has changed substantially since 1941: the lake level is much lower and the volume larger,
and the lake is directly connected to the steep glacierized slopes, so that the impact of a hypothetic landslide could be
very different now. Also, the current lake is dammed by another moraine than the pre-1941 lake, with a very different
dam geometry (Somos-Valenzuela et al., 2016). In general, the mechanisms of the landslide impact into the lake,
which were not the focus of the present study, would require more detailed investigations. Ideally, such work would
be based on a three-phase model (Pudasaini and Mergili, 2019; considering ice as a separate phase), and consider
knowledge and experience gained from comparable, well-documented events. A possible candidate for such an event
would be the 2010 event at Laguna 513, which was back-calculated by Schneider et al. (2014). Another remaining
issue is the lateral spreading of the flow on the fan of Huaráz, which is overestimated in all four simulations (Figs. 8, 9,
and 12): the most likely reason for this is the insufficient representation of fine-scale structures such as buildings or
walls in the DEM, which would serve as obstacles confining the flow in lateral direction.

## 7 Conclusions

We have performed back-calculations of the documented 1941 GLOF process chain involving Lake Palcacocha and
Lake Jircacocha in the Quilcay catchment in the Cordillera Blanca, Perú. The key messages of this work are summa-
rized as follows:
• Retrogressive erosion, possibly caused by a minor impact wave, appears to be the most likely release mecha-
nism of the process chain, facilitated by a geotechnically poorly stable dam with a low width-to-height ratio.
This type of failure – a combination of the idealized scenarios considered in this work – can be inferred from
observations, and appears most plausible with regard to the simulation results. The identification of the trig-
gering process remains difficult, also because the subsequent interaction with Lake Jircacocha disguises part of
the respective signature downstream.
• The correspondence between simulation results and observations is reasonable, and the model parameter val-
ues used are physically plausible. However, considerable uncertainties remain with regard to peaks and shapes
of the discharge hydrographs, and to the quantification of flow velocities and travel times. Adding a yield
strength to the fluid phase (Scenario AX) completely changes the temporal, but not the spatial evolution of the
flow. Still, travel times remain in the same order of magnitude as those derived by Somos-Valenzuela et al.
(2016) for possible future events.
• Transfer of the findings to forward simulations in the same area or elsewhere remains a challenge due to dif-
ferences in the initial conditions, uncertainties of the reference data, equifinality issues, and the effects of pro-
cess magnitude (Mergili et al., 2018b).



## Code availability

The model codes of r.avaflow, a manual, training data, and the necessary start scripts can be obtained from Mergili (2019).

## Data availability

The original DEM was provided by MINAM and may not be freely distributed, but all data derived within the present work can be obtained by directly contacting the first author (martin.mergili@boku.ac.at).

## Author contribution

MM developed the main ideas, defined the scenarios, did most of the data processing, simulations, and analyses, wrote the major portion of the text, and prepared all the figures and tables. SP provided important ideas with regard to the numerical simulations and contributed to the internal revision and optimization of the manuscript. AE contributed with important ideas, conducted field work, acquired data, contributed to the writing of the introductory chapters, and took part in the internal revision and optimization of the manuscript. JTF provided important contributions to the internal revision and optimization of the work. AC provided important data and contributed to the internal revision and optimization of the manuscript. HF contributed with important ideas and field work, data acquisition, and text blocks for the introductory chapters, and took part in the internal revision and optimization of the manuscript.

## Competing interests

The authors declare that they have no conflict of interest.

## Acknowledgements

Part of this work was conducted within the international cooperation project "A GIS simulation model for avalanche and debris flows (avaflow)" supported by the German Research Foundation (DFG, project number PU 386/3-1) and the Austrian Science Fund (FWF, project number I 1600-N30). Shiva P. Pudasaini further acknowledges financial support from DFG through the research project "A novel and unified solution to multi-phase mass flows: U_MultiSol". The work also follows the AKTION Austria – Czech Republic project "Currently forming glacial lakes: potentially hazardous entities in deglaciating high mountains" of Adam Emmer and Martin Mergili. Further, the support provided by the Swiss Agency for Development and Cooperation (SDC) through Proyecto Glaciares+, is acknowledged. Adam Emmer was also supported by the Ministry of Education, Youth and Sports of the Czech Republic within the National Sustainability Programme I (NPU I), grant number LO1415, and the postdoc grant of the Czech Academy of Sciences. Finally, we are grateful to Matthias Benedikt for comprehensive technical support in relation to r.avaflow.

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





## Tables

Table 1. Characteristics of Lake Palcacocha (1941 and 2016) and Lake Jircacocha (1941), and changes due to the 1941 GLOF. Topographic reconstruction according to field observations, historic photographs, Vilímek et al. (2005), ANA (2016).

| Parameter | Lake Palcacocha at 1941 GLOF | Lake Palcacocha 2016 | Lake Jircacocha at 1941 GLOF |
|---|---|---|---|
| Lake level elevation (m a.s.l.) | 4,610 | 4,563 | ~4,130 |
| Surface area ($10^3$ m²) | 303 | 514 | 215 |
| Lake volume ($10^6$ m³) | 12.9 [1] | 17.4 | 3.3 |
| GLOF volume ($10^6$ m³) | 10.9 [2] | – | 3.3 |
| Max. lake depth (m) | 108[3] | 71 | 33 |
| Lowering of lake level (m) | 47 [2] | – | 33 |

[1] Reference values differ among sources: according to Vilímek et al. (2005), the volume of Lake Palcacocha in 1941 was 9–11 million m³, whereas a reconstruction of ANA resulted in 13.1 million m³. In contrast, Vilímek et al. (2005) estimate a pre-failure volume of 4.8 million m³ for Lake Jircacocha, whereas, according to ANA, the volume was only 3.0 million m³.

[2] Computed from the difference between the pre-1941 lake level and the modern lake level (before mitigation) of 4563 m. A reconstruction of ANA in 1948 resulted in in a residual lake volume of approx. 100,000 m³ and a residual depth of 17 m, both much smaller than derived through the reconstruction in the present work. One of the reasons for this discrepancy might be the change of the glacier in the period 1941–1948.

[3] This value is highly uncertain and might represent an overestimation: the maximum depth of the lake strongly depends on the exact position of the glacier terminus, which was most likely located in an area of increasing lake depth in 1941.





Table 2. Reference information used for back-calculation of the 1941 process chain.

| Parameter | Value | Remarks | References |
|---|---|---|---|
| Impact area | 4.3 km² [1] | Mapped from post-event aerial images | Servicio Aerofoto-gramétrico Nacional |
| Breach volume – Palcacocha | 2.0 million m³ | Comparison of pre- and post-event DTMs | Topographic reconstruction ( Sect. 4) |
| Breach depth – Palcacocha | 76 m | Elevation change at reference point R1 (Fig. 4) | Topographic reconstruction (Sect. 4) |
| Breach volume – Jircacocha | 2.8 million m³ | Comparison of pre- and post-event DTMs | Topographic reconstruction (Sect. 4) |
| Material entrained upstream from Lake Jircacocha | 1.0 million m³ | Maximum, value might be much lower | Topographic reconstruction (Sect. 4) |
| Material entrained downstream from Lake Jircacocha | 3.1 million m³ | Maximum, value might be much lower | Topographic reconstruction ( Sect. 4) |
| Material entrained in promontory | 7.3 million m³ | Maximum, value might be much lower | Topographic reconstruction (Sect. 4) |
| Maximum depth of entrainment in promontory | 50 m | Rough estimate | Somos-Valenzuela et al. (2016) |
| Material arriving at Huaráz | 4–6 million m³ | | Kaser and Georges (2003) |

[1] Includes the surface of Lake Palcacocha





Table 3. Key model parameters applied to the simulations in the present work. Where three values are given, the first
value applies to the glacier, the second value to the remaining area upstream of the dam of Lake Jircacocha, and the
third value to the area downstream of the dam of Lake Jircacocha.

| Symbol | Parameter | Unit | Value |
| --- | --- | --- | --- |
| $\rho_S$ | Solid material density (grain density) | kg m$^{-3}$ | 2,700 |
| $\rho_F$ | Fluid material density | kg m$^{-3}$ | 1,000[1] |
| $\varphi$ | Internal friction angle | Degree | 28 |
| $\delta$ | Basal friction angle | Degree | 6, 12, 7 |
| $v$ | Kinematic viscosity of fluid | m² s$^{-1}$ | ~0 |
| $\tau_y$ | Yield strength of fluid | Pa | 0[2] |
| $C_{AD}$ | Ambient drag coefficient | – | 0.02, 0.005, 0.005 |
| $C_{FF}$ | Fluid friction coefficient | – | 0.001, 0.004, 0.004 |
| $C_E$ | Entrainment coefficient | – | $10^{-6.75}$[3], $10^{-7.15}$[4] |

[1] The fluid material density is set to 1,100 kg m$^{-3}$ in Scenario AX.
[2] The yield strength of the fluid phase is set to 5 Pa in Scenario AX.
[3] This value applies to the dam of Lake Palcacocha.
[4] This value applies to all other areas.





Table 4. Empirical relationships for the peak discharge in case of breach of moraine and landslide dams (Walder and
O'Connor, 1997), and the peak discharges estimated for Lake Palcacocha and Lake Jircacocha. $q_p$ = peak discharge
(m³ s⁻¹), $V$ = total volume of water passing through the breach (m³); $D$ = drop of lake level (m); REG = regression;
ENV = envelope. The values of $V$ and $D$ for the two lakes are summarized in Table 1. See also Rivas et al. (2015).

| Moraine | $a_{REG}$ | $a_{ENV}$ | $b$ | $q_p$ Palcacocha REG (m³ s⁻¹) | $q_p$ Palcacocha ENV (m³ s⁻¹) |
|---|---|---|---|---|---|
| $q_p = a \cdot V^b$ | 0.045 | 0.22 | 0.66 | 2,231 | 10,905 |
| $q_p = a \cdot D^b$ | 60.3 | 610 | 0.84 | 1,531 | 15,484 |
| $q_p = a \cdot (V \cdot D)^b$ | 0.19 | 1.1 | 0.47 | 2,560 | 14,819 |
| Landslide | $a_{REG}$ | $a_{ENV}$ | $b$ | $q_p$ Jircacocha REG (m³ s⁻¹) | $q_p$ Jircacocha ENV (m³ s⁻¹) |
| $q_p = a \cdot V^b$ | 1.6 | 46 | 0.46 | 1,638 | 47,101 |
| $q_p = a \cdot D^b$ | 6.7 | 200 | 1.73 | 2,839 | 84,734 |
| $q_p = a \cdot (V \cdot D)^b$ | 0.99 | 25 | 0.4 | 1,662 | 41,973 |






Table 5. Summary of the key results obtained with the computational experiments A–C. Refer to Tables 1 and 2 for the
volumes involved, and to Table 4 for empirically expected peak discharges. Note that all entrained volumes are com-
posed of 80% of solid and 20% of fluid material in terms of volume.

| Scenario | A | AX | B | C |
|---|---|---|---|---|
| Description | Overtopping | Overtopping | Impact wave | Dam collapse |
| Entrained volume Lake Palcacocha dam (m³) | 1.5 million | 1.4 million | 2.7 million | – |
| Fluid peak discharge at outlet of Lake Palcacocha (m³ s⁻¹) | 19,000 | 8,200 | 17,000[1] | 38,000 |
| Entrained volume Lake Jircacocha dam (m³) | 2.2 million | 2.0 million | 2.2 million | 2.2 million |
| Fluid peak discharge at outlet of Lake Jircacocha (m³ s⁻¹) | 14,700 | 7,600 | 15,000 | 15,400 |
| Material entrained up-stream from Lake Jircaco-cha (m³) | 0.7 million | 0.7 million | 0.7 million | 0.7 million |
| Material entrained down-stream from Lake Jircaco-cha (m³) | 1.5 million | 1.3 million | 1.5 million | 1.5 million |
| Material entrained in promontory (m³) | 5.3 million | 5.3 million | 5.3 million | 5.3 million |
| Travel time to Huaráz (s) Start (Peak) | 2,760 (3,660) | 4,200 (6,480) | 3,060 (4,080) | 2,160 (3,060) |
| Solid delivered to Huaráz (m³) | 2.5 million | 2.6 million | 2.5 million | 2.7 million |

[1] Peak of initial overtopping as response to the impact wave: 7,000 m³ s⁻¹



**Figures**

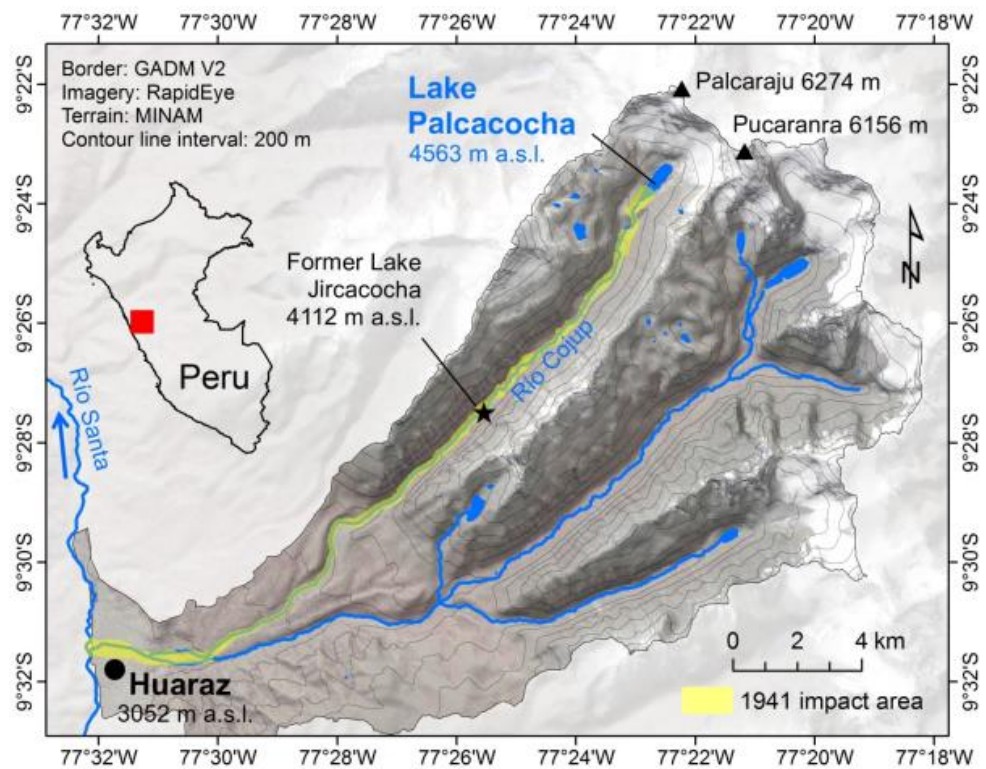

Fig. 1. Location and main geographic features of the Quilcay catchment with Lake Palcacocha and the former Lake Jircacocha.

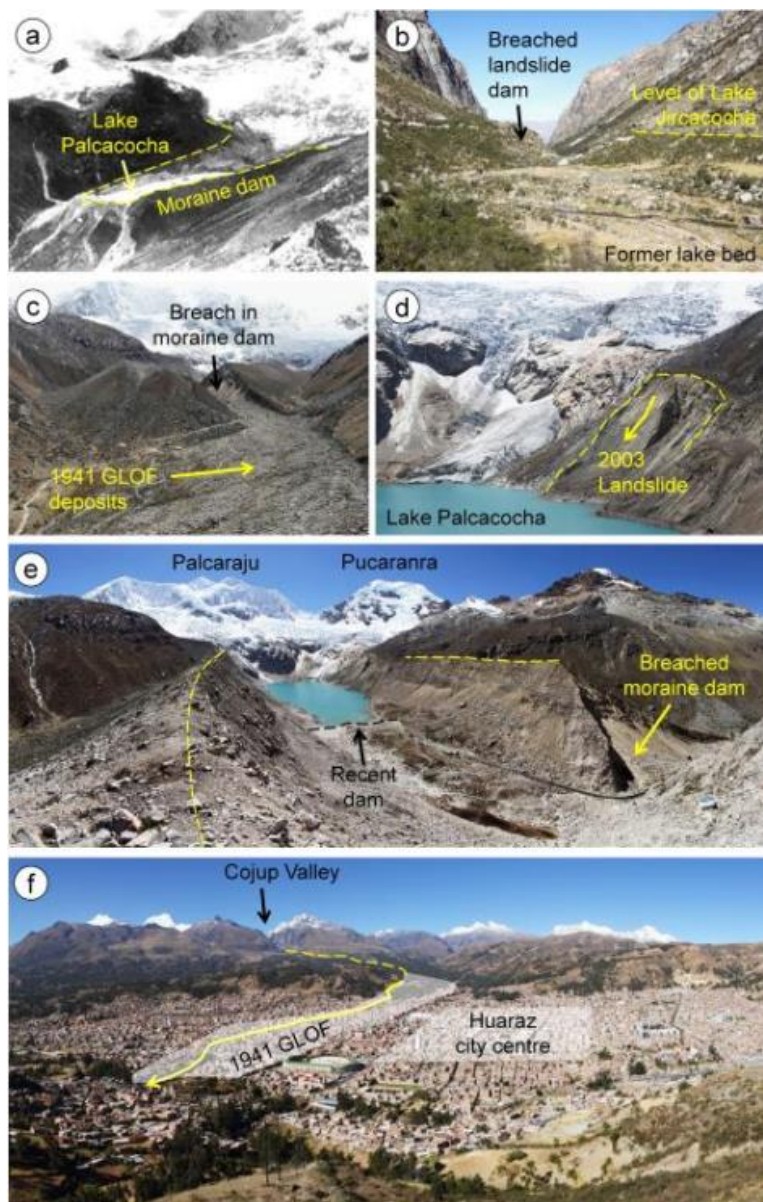

789

Fig. 2. The Quilcay catchment from Lake Palcacocha down to Huaráz. (a) Lake Palcacocha in 1939, two years prior to
the 1941 event. (b) The site of former Lake Jircacocha with the breached landslide dam and the former lake level. (c)
Breached moraine dam and 1941 GLOF deposits, seen from downstream. (d) Left lateral moraine of Lake Palcacocha
with landslide area of 2003. (e) Panoramic view of Lake Palcacocha, with the breach in the moraine dam and the
modern lake impounded by a smaller terminal moraine and two artificial dams. (f) Panoramic view of Huaráz with
city centre and approximate impact area of the 1941 event. Photos: (a) Hans Kinzl, 1939 (Kinzl and Schneider, 1950);
(b) Martin Mergili, July 2017; (c) Gisela Eberhard, July 2018; (d)–(f): Martin Mergili, July 2017.






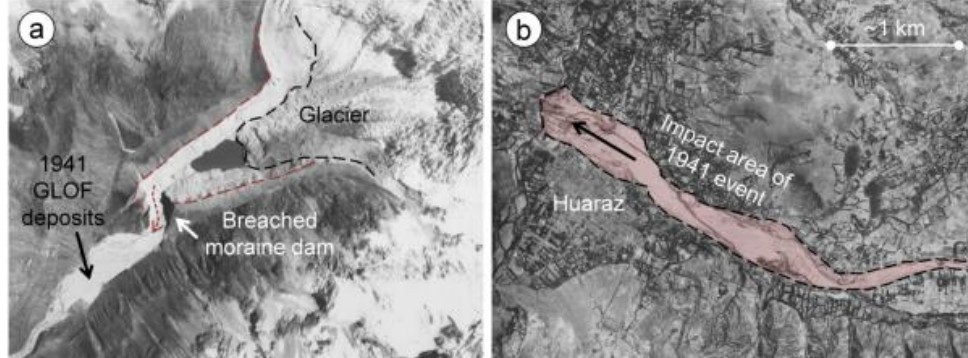

Fig. 3. Situation in 1948, seven years after the 1941 event. (a) Residual Lake Palcacocha, and traces of the 1941 event.
(b) Huaráz with the impact area of the 1941 event. Imagery source: Servicio Aerofotogramétrico Nacional, Perú.

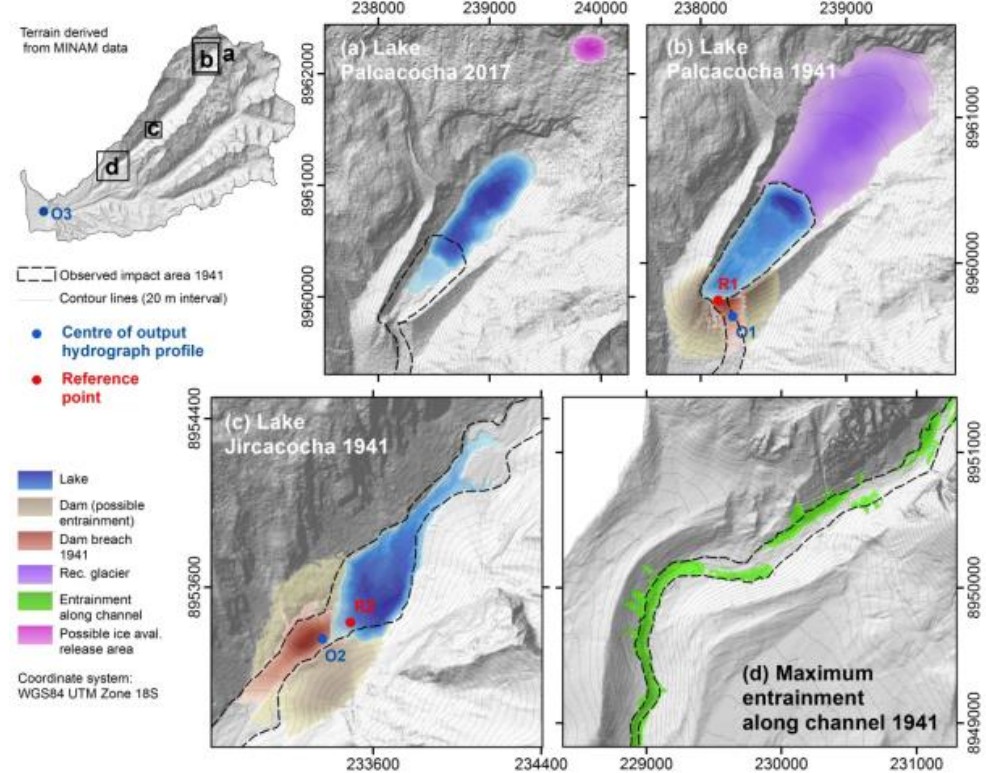

Fig. 4. Reconstruction of lakes and topography. (a) Lake Palcacocha in 2017. (b) Lake Palcacocha before the 1941 event. (c) Lake Jircacocha before the 1941 event. (d) Part of the promontory section of the Cojup Valley, with lowering of the valley bottom by up to 50 m. The possible rock avalanche release area is shown in (a) for clarity, but is applied to the 1941 situation.



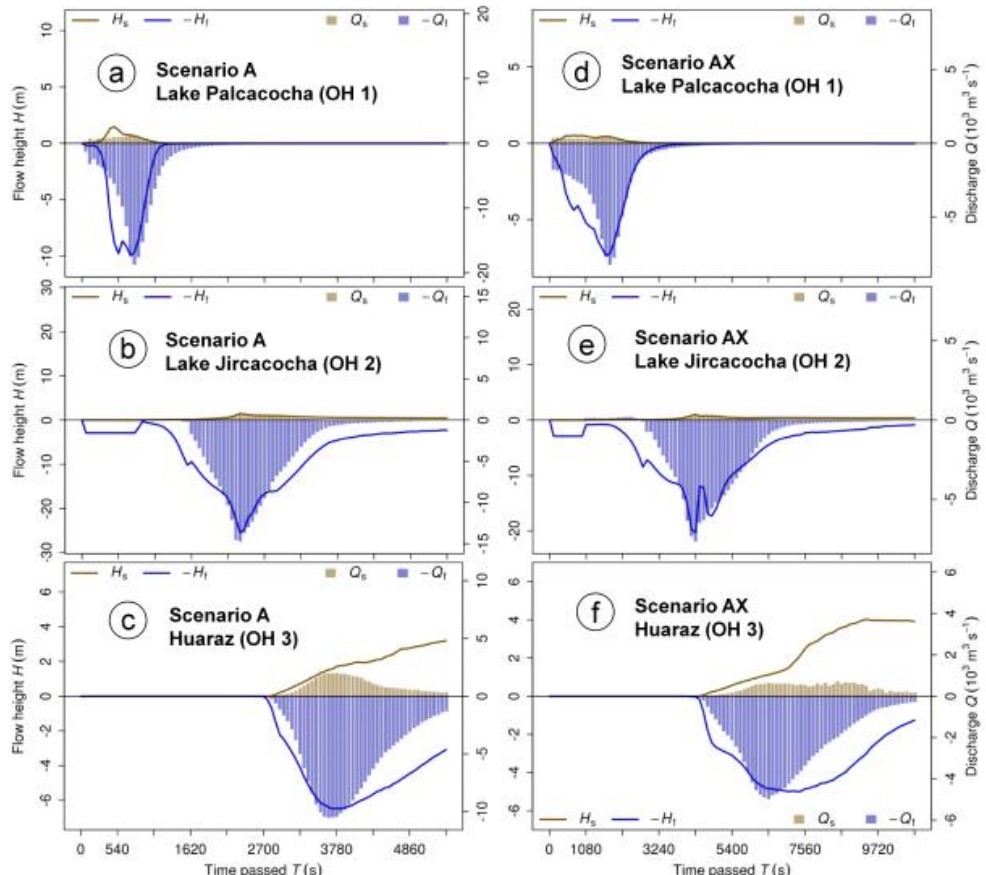

Fig. 5. Hydrographs of moraine dam failure of Lake Palcacocha (a, d), landslide dam failure of Lake Jircacocha (b, e), and the flow entering the urban area of Huaráz (c, f) for the scenarios A and AX. Note that, for clarity, fluid flow heights and discharges are plotted in negative direction.



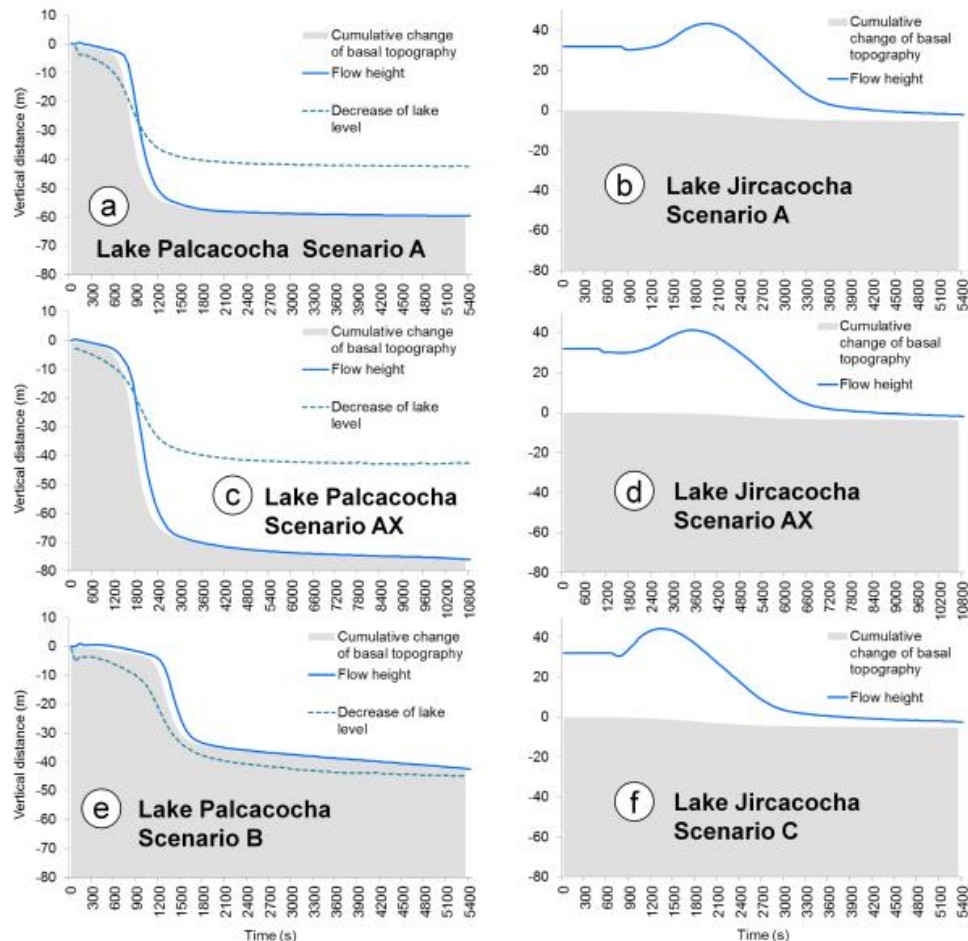


Fig. 6. Evolution of flow height and basal topography at the outlets of Lake Palcacocha (reference point R1 in Fig. 4b),
and Lake Jircacocha (reference point R2 in Fig. 4c). The reference points are placed in a way to best represent the evo-
lution of the breach in the dam for Lake Palcacocha, and the evolution of the impact wave for Lake Jircacocha. Addi-
tionally, the evolution of the lake level is shown for Lake Palcacocha. Note that the result for Scenario B is only dis-
played for Lake Palcacocha (e), whereas the result for Scenario C is only illustrated for Lake Jircacocha (f). The vertical
distance displayed on the y axis refers to the terrain height or the lake level at the start of the simulation, respectively,
whereby the flow height is imposed onto the topography. In Scenario B, the initial impact wave at the dam of Lake
Palcacocha is only poorly represented due to the low temporal resolution of the simulation, and due to blurring by
numerical effects (e).



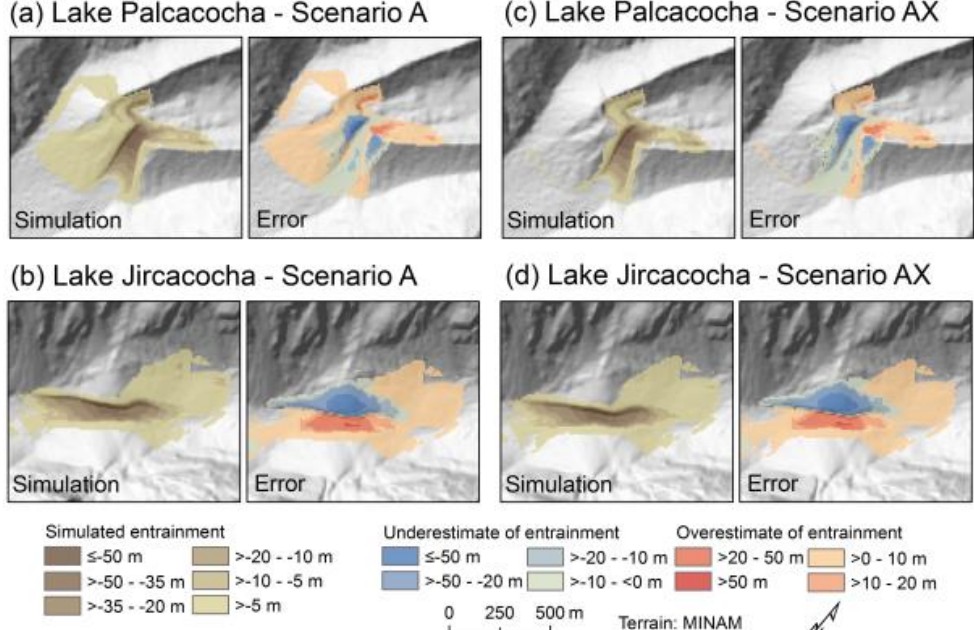

Fig. 7. Simulated versus reconstructed entrainment patterns for the scenarios A and AX. The total entrained height
and the difference between simulated and reconstructed entrainment (error) are shown. (a) Lake Palcacocha, Scenario
A. (b) Lake Jircacocha, Scenario A. (c) Lake Palcacocha, Scenario AX. (d) Lake Jircacocha, Scenario AX.





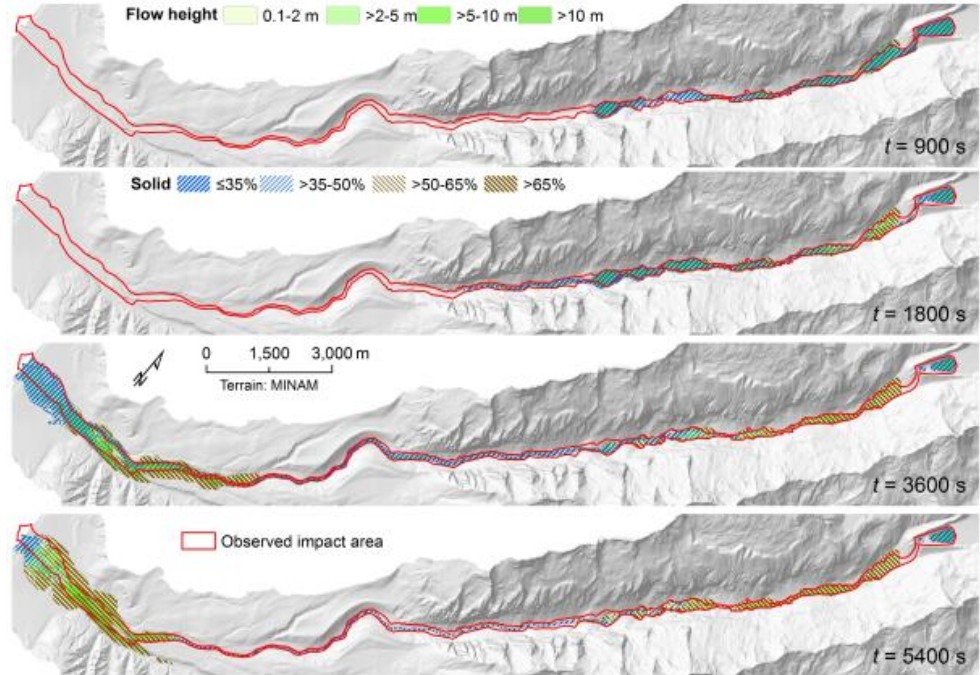


Fig. 8. Evolution of the flow in space and time (Scenario A).



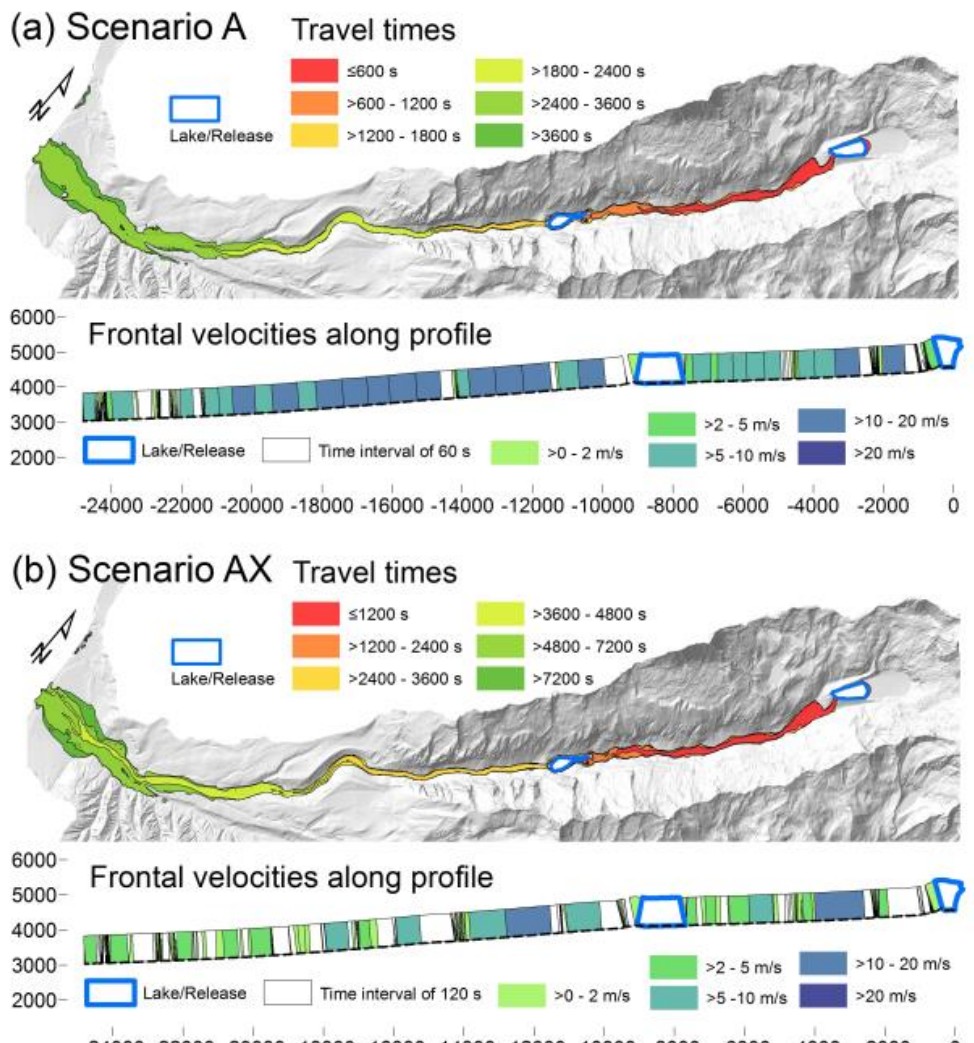

Fig. 9. Travel times and frontal velocities for the scenarios (a) A and (b) AX. Void fields in the profile graph refer to areas without clearly defined flow front.



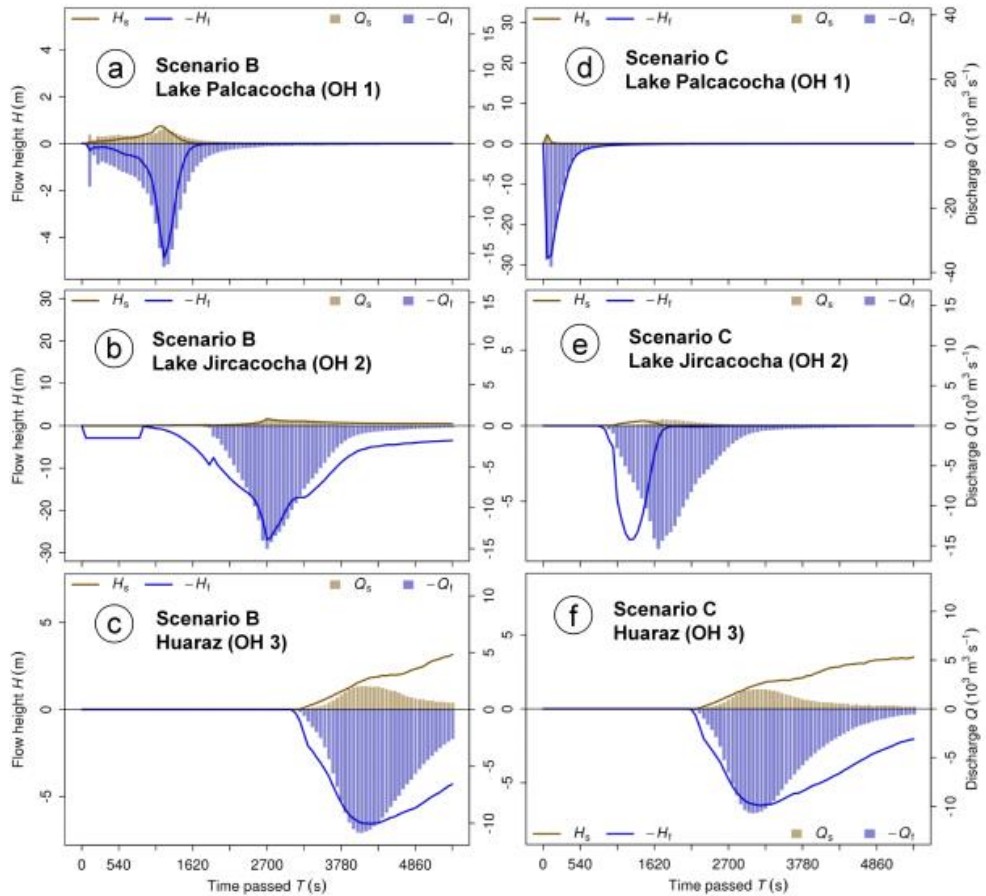

Fig. 10. Hydrographs of moraine dam failure of Lake Palcacocha (a, d), landslide dam failure of Lake Jircacocha (b, e),
and the flow entering the urban area of Huaráz (c, f) for the scenarios B and C. Note that, for clarity, fluid flow heights
and discharges are plotted in negative direction.





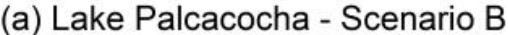

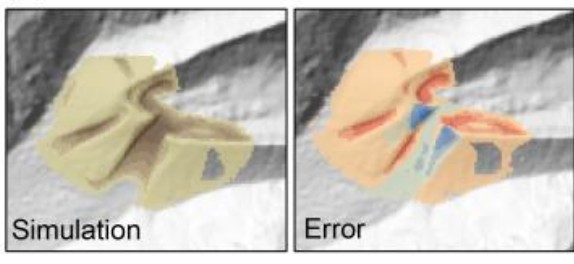

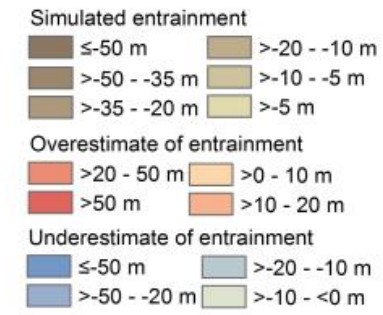

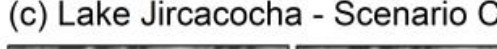

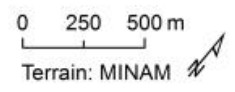

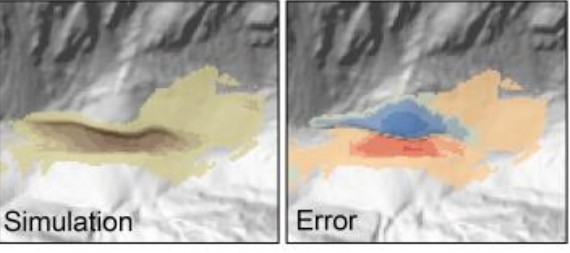

Fig. 11. Simulated versus reconstructed entrainment patterns for the scenarios B and C. The total entrained height and
the difference between simulated and reconstructed entrainment (error) are shown. (a) Lake Palcacocha, Scenario B.
(b) Lake Jircacocha, Scenario B. (c) Lake Jircacocha, Scenario C.





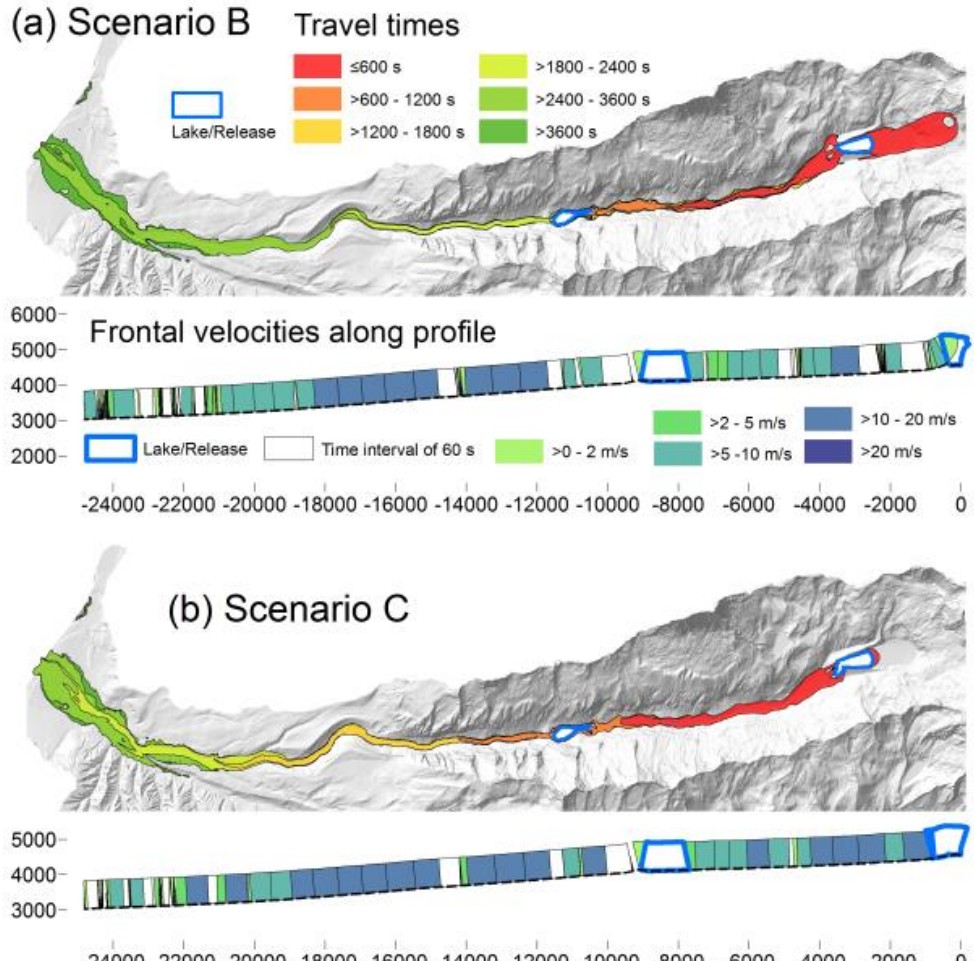

Fig. 12. Travel times and frontal velocities for the scenarios (a) B and (b) C. Note that the legend of (a) also applies to
(b). Void fields in the profile graph refer to areas without clearly defined flow front.