# Peer review of "Reconstruction of the 1941 GLOF process chain at Lake Palcacocha (Cordillera Blanca, Perú)"

_Hydrology and Earth System Sciences, 2019_

## Referee Comment (RC1) · John Reynolds (Referee) · 27 Sep 2019

This is an interesting and useful paper that serves two important functions. Firstly, it provides a back analysis of the 1941 GLOF process chain from Laguna Palcacocha, Peru, that helps an understanding of the physical processes associated with the event. Secondly, it is a useful demonstration or r.avaflow software that is fast becoming more widely applied for such studies.

The only comment I would make in this discussion is to expand a little on the detail behind the Laguna 513 situation between 1988 and 1994 as many papers that cite this lake do not report the full story and it merits telling again. Details of the remediation work undertaken have been described by Reynolds, J.M. 1998. Managing the risks of

glacial flooding at hydro plants. Hydro Review Worldwide, 6(2):18-22; and by Reynolds, J.M., Dolecki, A. and Portocarrero, C. 1998. The construction of a drainage tunnel as part of glacial lake hazard mitigation at Hualcán, Cordillera Blanca, Peru. In: Maund, J. & Eddleston, M. (eds.) Geohazards in engineering geology. Geological Society Engineering Group Special Publication No. 15, pp. 41-48.

In essence, in late 1988, surveying undertaken by local engineers identified that the small moraine dam impounding Lake 513 was ice cored and that this ice core was subsiding through ablation by ∼11 cm/month. It was a simple calculation, therefore, to estimate that by early 1989, the subsidence would have reduced the freeboard to zero and worse, would have resulted in the moraine dam failing and being eroded leading to an outburst flood. The local engineers, led by Cesar Portocarrero, identified that siphoning would be sufficient and practical to reduce the lake level by 3 m or so to alleviate a possible outburst. However, they had insufficient funds to purchase the necessary siphons. Two days before Christmas 1988, Cesar phoned me in the UK from Peru to ask if I could help. A few phone calls and several hours later I had managed to persuade the British Embassy in Lima to provide the necessary funds. Consequently, within a couple of weeks the siphons had been installed and the lake level lowered.

In 1991, a small ice avalanche, thought to have originated from the hanging glacier perched above the lake, fell into the lake producing a small displacement wave. However, this was sufficient to breach the remains of the moraine dam and produce a small outburst flood. It had the consequences of lowering the water level down to and exposed a solid rock bat that had been beneath the terminal moraine dam. The new water level was by this time only just below the rim of the rock bar.

Ing. Portocarrero began to design a more permanent mitigation scheme of tunnelling through the rock bar to lower the lake by 20 m. In 1993, having been informed of his design, it became apparent that the water hydrostatic pressure under 20 m plus head of water could rupture the discharge portal end of the proposed tunnel leading to a greater failure of the distal flank of the rock bar. With emergency funding provided by the British

Government, in late 1993 Reynolds and Dolecki visited the site operations with Ing. Portocarrero. We came up with a scheme for which the equipment was already on site that required the excavation of a tiered suite of tunnels whose inflow portals were set 5 m vertically apart, with the uppermost tunnel being opened first, to lower the lake level down by 5 m; then the second tunnel, for a further 5 m lowering. Explosives failed to detonate for the break through for the third tunnel, so it was decided to go for a 10 m breach through to the lowermost tunnel, which was established safely and the lake was successfully lowered by 20 m, thereby creating a freeboard against avalanche push waves and displacement waves in the case of further avalanche activity. The thinking at that time was that an ice avalanche would most probably originate from the ice cliff associated with the perched hanging glacier immediately above the upstream end of the lake.

The rock/ice avalanche that occurred in 2010 was from the uppermost flanks of the backwall above the lake. This was then when it was realised that this avalanche might have been triggered by thawing of permafrost where the rock face was exposed. Thankfully, having lowered the lake level by May 994 by 20 m, when the avalanche occurred in 2010, the exposed rock bar with 20 m of freeboard accommodated most of the 28-m high avalanche push wave, with only a residual amount overtopping the rock bar. Had the further remediation not have been undertaken, the consequences of this 2010 would most likely have been far more tragic, with possibly as many as 5-6,000 fatalities, as defined by the local mayor. Whilst the 2010 GLOF/alluvion caused damage, especially to the outskirts of the town, there were no casualties.

---

## Referee Comment (RC2) · Anonymous Referee #2 · 30 Sep 2019

The paper deals with r.avaflow-modelling of the GLOF process chain at Lake Palca-cocha, back in 1941. The topic is very important, as the model could be used to simulate future hazardous events at the investigated site. Hence, the paper is relevant and lies within the journal's scope. The following aspects could be addressed as well: - The detailed arguments for the four chosen scenarios are not completely clear and it would be very interesting to inform the reader with more details. - In addition, a sensitivity analysis would be of great interest and an explanation of the range of model errors. This could maybe also be an explanation for the scenarios. - The authors could give a clearer recommendation about the use of the current model for modelling future events. Since the input data is not clear, what about the model parameters to be used? This refers again to the sensitivity analysis and model errors.

---

## Referee Comment (RC3) · Ashim Sattar (Referee) · 8 Oct 2019

The manuscript addresses a relevant problem in glacial hazard studies. It is well written and presents some very interesting results of GLOF reconstruction. The study has significant scientific and practical value for understanding GLOF events in the past. It confirms the scope of the journal HESS an is fit for publication (few minor comments below). The fact that Cordillera Blanca has been showing rapid glacier recession over the past few decades, there is a great need to quantify the impact of such failure events in the past. Assessment of the GLOF hydraulics helps to evaluate the extremity in terms of damage, these events can cause to the downstream regions. The data produced can be helpful in the decision-making process to identify lakes with similar potential in the valley or its surroundings. Further, it demonstrates the application of open-source

mass flow simulation (r.avaflow) to numerically back-calculate a historical GLOF event (of Lake Palcacocha) and its cascading effect on Lake Jircacocha (landslide barrier lake). The methods are clearly outlined in the manuscript. The results produced in the study is sufficient to support the interpretations and conclusions. However, the discussion section lacks a comparative analysis, the results do not show any quantitative comparison with other studies in the region (eg. Laguna 513). Overall, it is a very comprehensive and well-written manuscript.

Few minor comments:

1. Line 42-45- I will suggest to include the latest literature here. Several GLOF impact modeling studies have been carried out in the Himalaya recently (2018-19).

2. The abstract is too general and does not reflect the specific quantitative results. Text in the abstract (line 23-24) can be shortened and instead information about the results can be included.

Figures:

Figure 1-The number of lat/long labels can be reduced

Figure 2 (f)-The impact area ends very abruptly. This is surprising. The inundation zone can be rechecked.

Figure 12 (a and b)- The terrain ends abruptly towards the downstream region (left corner); a small patch of inundation boundary is visible (top left corner), kindly recheck.

---

## Author Comment (AC2) · 4 Nov 2019

Please see supplement.

Please also note the supplement to this comment:
https://www.hydrol-earth-syst-sci-discuss.net/hess-2019-357/hess-2019-357-AC2-supplement.pdf

---

## Author Response (AR1)

**Reconstruction of the 1941 GLOF process chain at Lake Palcacocha (Cordillera Blanca, Perú)**

Martin Mergili, Shiva P. Pudasaini, Adam Emmer, Jan-Thomas Fischer, Alejo Cochachin, and Holger Frey

**Response to the comments of Referee #1 (John Reynolds)**

We would like to thank the reviewer for the constructive remarks. Below, we address each comment in full detail. Our response is written in blue colour. Changes in the manuscript are highlighted in yellow colour.

This is an interesting and useful paper that serves two important functions. Firstly, it provides a back analysis of the 1941 GLOF process chain from Laguna Palcacocha, Peru, that helps an understanding of the physical processes associated with the event. Secondly, it is a useful demonstration or r.avaflow software that is fast becoming more widely applied for such studies. The only comment I would make in this discussion is to expand a little on the detail behind the Laguna 513 situation between 1988 and 1994 as many papers that cite this lake do not report the full story and it merits telling again. Details of the remediation work undertaken have been described by *Reynolds, J.M. 1998. Managing the risks of glacial flooding at hydro plants. Hydro Review Worldwide, 6(2):18-22*; and by *Reynolds, J.M., Dolecki, A. and Portocarrero, C. 1998. The construction of a drainage tunnel as part of glacial lake hazard mitigation at Hualcán, Cordillera Blanca, Peru. In: Maund, J. & Eddleston, M. (eds.) Geohazards in engineering geology. Geological Society Engineering Group Special Publication No. 15, pp. 41-48*.

In essence, in late 1988, surveying undertaken by local engineers identified that the small moraine dam impounding Lake 513 was ice cored and that this ice core was subsiding through ablation by~11 cm/month. It was a simple calculation, therefore, to estimate that by early 1989, the subsidence would have reduced the freeboard to zero and worse, would have resulted in the moraine dam failing and being eroded leading to an outburst flood. The local engineers, led by Cesar Portocarrero, identified that siphoning would be sufficient and practical to reduce the lake level by 3 m or so to alleviate a possible outburst. However, they had insufficient funds to purchase the necessary siphons. Two days before Christmas 1988, Cesar phoned me in the UK from Peru to ask if I could help. A few phone calls and several hours later I had managed to persuade the British Embassy in Lima to provide the necessary funds. Consequently, within a couple of weeks the siphons had been installed and the lake level lowered. In 1991, a small ice avalanche, thought to have originated from the hanging glacier perched above the lake, fell into the lake producing a small displacement wave. However, this was sufficient to breach the remains of the moraine dam and produce a small outburst flood. It had the consequences of lowering the water level down to and exposed a solid rock bat that had been beneath the terminal moraine dam. The new water level was by this time only just below the rim of the rock bar. Ing. Portocarrero began to design a more permanent mitigation scheme of tunneling through the rock bar to lower the lake by 20 m. In 1993, having been informed of his design, it became apparent that the water hydrostatic pressure under 20 m plus head of water could rupture the discharge portal end of the proposed tunnel leading to a greater failure of the distal flank of the rock bar. With emergency funding provided by the British Government, in late 1993 Reynolds and Dolecki visited the site operations with Ing. Portocarrero. We came up with a scheme for which the equipment was already on site that required the excavation of a tiered suite of tunnels whose inflow portals were set 5 m vertically apart, with the uppermost tunnel

being opened first, to lower the lake level down by 5 m; then the second tunnel, for a further 5 m lowering. Explosives failed to detonate for the break through for the third tunnel, so it was decided to go for a 10m breach through to the lowermost tunnel, which was established safely and the lake was successfully lowered by 20 m, thereby creating a freeboard against avalanche push waves and displacement waves in the case of further avalanche activity. The thinking at that time was that an ice avalanche would most probably originate from the ice cliff associated with the perched hanging glacier immediately above the upstream end of the lake. The rock/ice avalanche that occurred in 2010 was from the uppermost flanks of the back wall above the lake. This was then when it was realised that this avalanche might have been triggered by thawing of permafrost where the rock face was exposed. Thankfully, having lowered the lake level by May 994 by 20 m, when the avalanche occurred in 2010, the exposed rock bar with 20 m of freeboard accommodated most of the 28-m high avalanche push wave, with only a residual amount overtopping the rock bar. Had the further remediation not have been undertaken, the consequences of this 2010 would most likely have been far more tragic, with possibly as many as 5-6,000 fatalities, as defined by the local mayor. Whilst the 2010 GLOF/alluvion caused damage, especially to the outskirts of the town, there were no casualties.

We are very glad to see that the reviewer likes our paper. We have included the 2010 event at Laguna 513 in the introduction and the discussion of the revised manuscript, referring to the suggested literature. We have kept the text blocks concerning Laguna 513 brief and concise, clearly relating them to those aspects also relevant for the present work. A more detailed account of the remediation measures and the 2010 event would be out of scope here, but could be of great interest for a future study. In the revised manuscript, we have mainly added the following pieces of text:

**Introduction (L71-73):**

Most notably, lowering the lake level of Laguna 513 through a system of tunnels in the 1990s has probably prevented a disaster downstream when a rock-ice avalanche impacted that lake in 2010 (Reynolds, 1998; Reynolds et al., 1998; Schneider et al., 2014).

**Discussion (L545-550):**

In principle, such an understanding can be transferred to present hazardous situations in order to inform the design of technical remediation measures. Earlier, measures were not only implemented at Lake Palcacocha (Portocarrero, 2014), but also at various other lakes such as Laguna 513: a tunnelling scheme implemented in the 1990s strongly reduced the impacts of the 2010 GLOF process chain (Reynolds, 1998; Reynolds et al., 1998; Schneider et al., 2014).

**Reconstruction of the 1941 GLOF process chain at Lake Palcacocha (Cordillera Blanca, Perú)**

Martin Mergili, Shiva P. Pudasaini, Adam Emmer, Jan-Thomas Fischer, Alejo Cochachin, and Holger Frey

**Response to the comments of Referee #2**

We would like to thank the reviewer for the constructive remarks. Below, we address each comment in full detail. Our response is written in blue colour. Changes in the manuscript are highlighted in yellow colour.

The paper deals with r.avaflow-modelling of the GLOF process chain at Lake Palcacocha, back in 1941. The topic is very important, as the model could be used to simulate future hazardous events at the investigated site. Hence, the paper is relevant and lies within the journal's scope. The following aspects could be addressed as well:

- The detailed arguments for the four chosen scenarios are not completely clear and it would be very interesting to inform the reader with more details.

Those arguments are elaborated at the end of the introduction of the discussion paper and, particularly, at the beginning of the discussion, but we agree that they should also be at least briefly explained In the place where each scenario is described. Therefore, in the revised manuscript, we have added the following statement (L287-291):

As the trigger of the sudden drainage of Lake Palcacocha is not clear, we consider four scenarios, based on the situation before the event as shown in the photo taken by Hans Kinzl, experiences from other documented GLOF events in the Cordillera Blanca (Schneider et al., 2014; Mergili et al., 2018a), considerations by Vilímek et al. (2005), Portocarrero (2014), and Somos-Valenzuela et al. (2016), as well as geotechnical considerations:

- In addition, a sensitivity analysis would be of great interest and an explanation of the range of model errors. This could maybe also be an explanation for the scenarios.

This is a very good point. We agree that a detailed, systematic sensitivity analysis could provide additional insight in some challenges of model parameterization, and help to quantify model errors. However, we have deliberately decided not to show a detailed sensitivity analysis for the following reasons:

- The paper is intended to tell the story about the 1941 Palcacocha event rather than about r.avaflow. Adding a sensitivity analysis, and describing and discussing it at an appropriate level of detail, would shift the paper much more in a technical direction, which is something we would like to avoid.
- This case study is not ideal for a sensitivity analysis due to long computational times (very long travel distance, therefore a very large number of raster cells to be processed). We have performed sensitivity analyses and parameter studies in earlier publications on r.avaflow (Mergili et al. 2018a, b) and most findings from those papers are most likely valid also for this work.

Therefore, in the revised manuscript, we consider the findings of those previous studies in more detail, and relate them to the 1941 event at Lake Palcacocha, rather than to directly perform another systematic sensitivity analysis.

Consequently, we have added the following text to the discussion of the revised manuscript (L562-575):

In general, it remains a challenge to reliably predict the outcomes of given future scenarios. The magnitude of the 1941 event was amplified by the interaction with Lake Jircacocha, whereas the 2012 GLOF process chain in the Santa Cruz Valley (Mergili et al., 2018a) alleviated due to the interaction with Lake Jatuncocha, comparable in size. While it seems clear that the result of such an interaction depends on event magnitude, topography, and the dam characteristics of the impacted lake, Mergili et al. (2018a, b) have demonstrated the high sensitivity of the behaviour of the simulated flow to the friction parameters, but also to the material involved (release mass, entrainment). A larger number of back-calculated process chains will be necessary to derive guiding parameter sets which could facilitate predictive simulations, and so will an appropriate consideration of model uncertainties and possible threshold effects (Mergili et al., 2018b). Earlier studies, considering the 2010 event at Laguna 513 (Schneider et al., 2014) and three future scenarios for Lake Palcacocha (Somos-Valenzuela et al., 2016) have followed a different strategy, using model cascades instead on integrated simulations, so that a comparison with studies based on r.avaflow is only possible to a limited extent.

- The authors could give a clearer recommendation about the use of the current model for modelling future events. Since the input data is not clear, what about the model parameters to be used? This refers again to the sensitivity analysis and model errors.

Yes, we have included this issue more prominently in the discussion of the revised manuscript (see also response to Referee #3 and the response to the previous comment, which is closely related to this one).

**Reconstruction of the 1941 GLOF process chain at Lake Palcacocha (Cordillera Blanca, Perú)**

Martin Mergili, Shiva P. Pudasaini, Adam Emmer, Jan-Thomas Fischer, Alejo Cochachin, and Holger Frey

**Response to the comments of Referee #3 (Ashim Sattar)**

We would like to thank the reviewer for the constructive remarks. Below, we address each comment in full detail. Our response is written in blue colour. Changes in the manuscript are highlighted in yellow colour.

The manuscript addresses a relevant problem in glacial hazard studies. It is well written and presents some very interesting results of GLOF reconstruction. The study has significant scientific and practical value for understanding GLOF events in the past. It confirms the scope of the journal HESS an is fit for publication (few minor comments below). The fact that Cordillera Blanca has been showing rapid glacier recession over the past few decades, there is a great need to quantify the impact of such failure events in the past. Assessment of the GLOF hydraulics helps to evaluate the extremity in terms of damage, these events can cause to the downstream regions. The data produced can be helpful in the decision-making process to identify lakes with similar potential in the valley or its surroundings. Further, it demonstrates the application of open-source mass flow simulation (r.avaflow) to numerically back-calculate a historical GLOF event (of Lake Palcacocha) and its cascading effect on Lake Jircacocha (landslide barrier lake). The methods are clearly outlined in the manuscript. The results produced in the study is sufficient to support the interpretations and conclusions. However, the discussion section lacks a comparative analysis, the results do not show any quantitative comparison with other studies in the region (eg. Laguna 513). Overall, it is a very comprehensive and well-written manuscript.

We are very glad to see that the reviewer likes our manuscript. The suggestion to also refer to other events (Laguna 513) goes in the same direction as the suggestion made by Referee #1. We fully agree that such a discussion can add value to the paper, and have included it accordingly. However, we have done this in a qualitative rather than in a quantitative way, mainly highlighting the differences with other studies considering both the type of approach, and the results: to date, only the 2012 GLOF process chain in the Santa Cruz Valley has been analyzed with a two-phase model (Mergili et al., 2018a), whereas other studies (Schneider et al., 2014 for Laguna 513 and Somos-Valenzuela et al., 2016 for future scenarios of a Lake Palcacocha GLOF) were based on "model cascades" – adding a quantitative comparison would, in our opinion, be very difficult since the events and, partly, also the modelling approaches differ among themselves, and would shift the scope of the study. However, it could be a very interesting future direction.

We have added the following text to the discussion of the revised manuscript (L562-575):

In general, it remains a challenge to reliably predict the outcomes of given future scenarios. The magnitude of the 1941 event was amplified by the interaction with Lake Jircacocha, whereas the 2012 GLOF process chain in the Santa Cruz Valley (Mergili et al., 2018a) alleviated due to the interaction with Lake Jatuncocha, comparable in size. While it seems clear that the result of such an interaction depends on event magnitude, topography, and the dam characteristics of the impacted lake, Mergili et al. (2018a, b) have demonstrated the high sensitivity of the behaviour of the simulated flow to the friction parameters, but also to the material involved (release mass, entrainment). A larger number of backcalculated process chains will be necessary to derive guiding parameter sets which could facilitate predictive simulations, and so will an appropriate consideration of model uncertainties and possible threshold effects (Mergili et al., 2018b). Earlier studies, considering the 2010 event at Laguna 513 (Schneider et al., 2014) and three future scenarios for Lake Palcacocha (Somos-Valenzuela et al., 2016) have followed a different strategy, using model cascades instead on integrated simulations, so that a comparison with studies based on r.avaflow is only possible to a limited extent.

Few minor comments:

1. Line 42-45- I will suggest to include the latest literature here. Several GLOF impact modeling studies have been carried out in the Himalaya recently (2018-19).

Thank you very much for this remark – the following references have been included in the revised manuscript:

Sattar, A., Goswami, A., & Kulkarni, A. V. (2019a). Application of 1D and 2D hydrodynamic modeling to study glacial lake outburst flood (GLOF) and its impact on a hydropower station in Central Himalaya. Natural Hazards, 97(2), 535-553.

Sattar, A., Goswami, A., & Kulkarni, A. V. (2019b). Hydrodynamic moraine-breach modeling and outburst flood routing-A hazard assessment of the South Lhonak lake, Sikkim. Science of the Total Environment, 668, 362-378.

*Turzewski, M. D., Huntington, K. W., & LeVeque, R. J. (2019). The geomorphic impact of outburst floods: Integrating observations and numerical simulations of the 2000 Yigong flood, eastern Himalaya. Journal of Geophysical Research: Earth Surface.*

2. The abstract is too general and does not reflect the specific quantitative results. Text in the abstract (line 23-24) can be shortened and instead information about the results can be included.

We have shortened the general part of the abstract (L23–26 in the original manuscript) and included some fundamental information about the results (L32-34):

Most simulation scenarios indicate travel times between 36 and 70 minutes to reach Huaráz, accompanied with peak discharges above  $10,000 \text{ m}^3/\text{s}$ .

and also about the implications (L37-39):

Predictive simulations of possible future events have to be based on a larger set of back-calculated GLOF process chains, taking into account the expected parameter uncertainties and appropriate strategies to deal with critical threshold effects.

Figures:

**Figure 1-The number of lat/long labels can be reduced**

We have increased the interval between the tick marks and labels from two to four minutes.

Figure 2 (f)-The impact area ends very abruptly. This is surprising. The inundation zone can be rechecked.

The reason for this abrupt ending is that (i) the valley is bounded by a steep slope at this side, and (ii) part of the lowermost portion of the inundation zone is hidden behind the hillslope in the left foreground of the photo. We have tried to better indicate this in the figure and also indicated it in the figure caption in the revised manuscript.

Figure 12 (a and b)- The terrain ends abruptly towards the downstream region (left corner); a small patch of inundation boundary is visible (top left corner), kindly recheck.

Yes, the reason for this pattern is that a small part of the simulated flow has proceeded downstream the Río Santa Valley near the edge of the area of interest, instead of leaving the area of interest. This is an "edge effect" not considered a significant result of the study. Therefore, we have masked out this area in the revised Fig. 12, and also in the revised Fig. 9, where a similar effect is visible.

**Reconstruction of the 1941 GLOF process chain at Lake Palcaco-**

**2 cha (Cordillera Blanca, Perú)**

**Martin Mergili1,2, Shiva P. Pudasaini3, Adam Emmer4, Jan-Thomas Fischer5, Alejo Cochachin6, and Holger Frey7**

[revised manuscript text omitted]

tion ( Sect. 4) |  |  |  |
| Breach depth – Palca-
cocha                           | 76 m                       | Elevation change at ref-
erence point R1 (Fig. 4) | Topographic reconstruc-
tion (Sect. 4)  |  |  |  |
| Breach volume – Jirca-
cocha                          | 2.8 million m 3 | Comparison of pre- and post-event DTMs               | Topographic reconstruc-
tion (Sect. 4)  |  |  |  |
| Material entrained
upstream from Lake
Jircacocha   | 1.0 million m 3 | Maximum, value might
be much lower                | Topographic reconstruc-
tion (Sect. 4)  |  |  |  |
| Material entrained
downstream from Lake
Jircacocha | 3.1 million m 3 | Maximum, value might
be much lower                | Topographic reconstruc-
tion ( Sect. 4) |  |  |  |
| Material entrained in promontory                         | 7.3 million m 3 | Maximum, value might
be much lower                | Topographic reconstruc-
tion (Sect. 4)  |  |  |  |
| Maximum depth of
entrainment in prom-
ontory       | 50 m                       | Rough estimate                                       | Somos-Valenzuela et al.
(2016)          |  |  |  |
| Material arriving at
Huaráz                           | 4–6 million m 3 |                                                      | Kaser and Georges (2003)                   |  |  |  |
| 1) Includes the surface of Lake Palcacocha    |                            |                                                      |                                            |  |  |  |

[revised manuscript text omitted]

922 1) Peak of initial overtopping as response to the impact wave: 7,000 m3 s-1

**923 Figures**